https://doi.org/10.1038/s42003-022-04080-7　　**OPEN**
# Identifying behavioral structure from deep variational embeddings of animal motion

Kevin Luxem[1,2], Petra Mocellin[1,2], Falko Fuhrmann[1,2], Johannes Kürsch[1,2], Stephanie R. Miller[3,4], Jorge J. Palop[3,4], Stefan Remy ● [1,2,5,6,7 ✉] & Pavol Bauer[1,2,7]

Quantification and detection of the hierarchical organization of behavior is a major challenge in neuroscience. Recent advances in markerless pose estimation enable the visualization of high-dimensional spatiotemporal behavioral dynamics of animal motion. However, robust and reliable technical approaches are needed to uncover underlying structure in these data and to segment behavior into discrete hierarchically organized motifs. Here, we present an unsupervised probabilistic deep learning framework that identifies behavioral structure from deep variational embeddings of animal motion (VAME). By using a mouse model of beta amyloidosis as a use case, we show that VAME not only identifies discrete behavioral motifs, but also captures a hierarchical representation of the motif's usage. The approach allows for the grouping of motifs into communities and the detection of differences in community-specific motif usage of individual mouse cohorts that were undetectable by human visual observation. Thus, we present a robust approach for the segmentation of animal motion that is applicable to a wide range of experimental setups, models and conditions without requiring supervised or a-priori human interference.

---

[1] Leibniz Institute for Neurobiology (LIN), Department of Cellular Neuroscience, Magdeburg, Germany. [2] German Center for Neurodegenerative Diseases (DZNE), Bonn, Germany. [3] Gladstone Institute of Neurological Disease, San Francisco, CA 94158, USA. [4] Department of Neurology, University of California, San Francisco, San Francisco, CA 94158, USA. [5] Center for Behavioral Brain Sciences (CBBS), Magdeburg, Germany. [6] German Center for Mental Health (DZPG), Magdeburg, Germany. [7] These authors jointly supervised this work: Stefan Remy, Pavol Bauer. ✉email: stefan.remy@lin-magdeburg.de

The brain is a dynamical system and its dynamics are reflected in the actions it performs. Thus, observable motion is a valuable resource for understanding brain function. In most of the current neuroethological studies this resource has only been partially utilized[1]. Reaching the goal of maximizing information content requires a complete capture of observable motion and unbiased interpretation of behavioral complexity. Unsupervised methods provide a gateway for this purpose as they do not rely on human annotations like their counterparts, supervised methods[2–4]. Moreover, unsupervised methods are able to learn rich dynamical representations of behavior on a sub-second scale, which are otherwise not detectable[5–10]. The need for unsupervised behavioral quantification methods has been recently recognized and innovative approaches in this direction have been introduced[7,11,12]. While there is a broad agreement among researchers in computational ethology that observable behavior can be encoded in a lower dimensional subspace or manifold[7–10], current methods insufficiently capture the complete spatiotemporal dynamics of behavior[10].

Recently, pose estimation tools such as DeepLabCut (DLC)[13], SLEAP[14] and DeepPoseKit[15] enabled efficient tracking of animal body-parts via supervised deep learning. The robustness of deep neural networks allows for a high degree of generalization between datasets[13]. However, while such tools provide a continuous virtual marker signal of the animal body motion, the extraction of underlying dynamics and motifs remains a key challenge.

To address this challenge and provide a reliable and robust solution, we here developed Variational Animal Motion Embedding (VAME), an unsupervised probabilistic deep learning framework for discovery of underlying latent states in behavioral signals obtained from pose estimation tools or dimensionality reduced video information[16]. The input signal is learned and embedded into a lower dimensional space via a variational recurrent neural network autoencoder. Given the low dimensional representation, a Hidden-Markov-Model (HMM) learns to infer hidden states, which represent behavioral motifs. A major advantage of VAME is the ability to learn a disentangled representation of latent factors via the variational autoencoder (VAE) framework[17–19]. This allows the model to embed a complex data distribution into a simpler prior distribution, where segments of the behavioral signal are grouped by their spatiotemporal similarity. We use a powerful autoregressive encoder to disentangle latent factors of the input data. Our approach is inspired by recent advances in the field of temporal action segmentation[20], representation learning[21–24] and unsupervised learning of multivariate time series[25,26].

In this manuscript, we introduce the VAME model and workflow based on behavioral data obtained from a bottom-up recording system in an open-field. We demonstrate the capability of VAME to identify the motif structure, the hierarchical connection of motifs and their transitions in a use case of Alzheimer transgenic mice[27]. Within this use case, VAME is capable of detecting differences between the transgenic and control group, while no differences were detectable by human observers. In addition, we compare VAME to current state-of-the-art methods[7,11] on a benchmark dataset. Finally, we present a complete guide for users to adopt our framework, from the installation process to training a model and common pitfalls. Hence, we provide a self-contained manuscript for state-of-the-art computational ethology analysis.

## Results

**VAME: variational animal motion embedding.** In our experimental setup, we let mice move freely in an open-field arena (Fig. 1a). During the experiment the movement was continuously monitored with a bottom-up camera for 25 min ($N = 90,000$ frames). A major advantage of the bottom-up perspective is that it reveals most of the animal's kinematic with only one camera view, which can be efficiently tracked by pose estimation. Our goal is to build a model that can learn the behavioral structure purely from the kinematic pose tracking. In order to identify the postural dynamics of the animal from the video recordings we used DLC[13]. For tracking, we used six virtual markers, which were positioned on all four paws of the animal, the nose, and the tailbase (Fig. 1a, b)). We aligned the animal from its allocentric arena coordinates to its egocentric coordinates. For this, each frame was rotated around the center between nose and tail, so that the animal was aligned from left (tailbase) to right (nose) (Fig. 1a, c)). This resulted in a time-dependent series $\mathbf{X} \in \mathbb{R}^{N \times m}$ for each animal with $m = 10$ (x, y)-marker positions that captured the kinematic of specified body parts (see Methods for guidance on the VAME preprocessing and alignment functionality).

Our aim was to extract useful information from the time series data, allowing for an effective behavioral quantification given spatial and temporal information of body dynamics. For this, trajectory samples $\mathbf{x}_i \in \mathbb{R}^{m \times w}$ that are pre defined time windows (with length $w = 30$) (Fig. 1a, d)) were randomly sampled from $\mathbf{X}$ and represented the input to train the VAME model. Our first goal was to identify behavioral motifs, which we defined according to[10] as "stereotyped and re-used units of movements". The second goal was the identification of the hierarchical and transition structure of motifs aiming at the detection of patterns within these transitions.

The VAME model consists of three bidirectional recurrent neural networks (biRNN)[28] with gated recurrent units[29] as basic building blocks (Fig. 1a, e)). In our model, the encoder biRNN receives a trajectory sample $\mathbf{x}_i$ (i.e., 500 ms of behavior) and embeds the relevant information into a lower dimensional latent space $\mathbf{Z} \in \mathbb{R}^{d \times N-w}$. Learning is achieved by mapping $\mathbf{x}_i$ to a fixed vector representation $\mathbf{z}_i \in \mathbb{R}^d$ (where $d < m \times w$) and passing this onto a biRNN decoder, which decodes the lower dimensional vector into an approximation $\tilde{\mathbf{x}}_i$ of the input trajectory. Additionally, another biRNN decoder learns to anticipate the structure of the subsequent time series trajectory $\tilde{\mathbf{x}}_{i+1} \in \mathbb{R}^{m \times v}$ from $\mathbf{z}_i$, thereby regularizing $\mathbf{Z}$ and forcing the encoder to learn improved dynamical features from the behavioral time series[30]. Here, $v$ is the prediction time window. We tested different model choices including single decoder model (only reconstruction) in Supplementary Section 1 and showed that a two decoder model (reconstruction and prediction) has improved performance on our tested metrics (see Methods).

The model is trained as variational autoencoder (VAE)[17] with a standard normal prior (see Methods "Variational animal motion embedding" for details on the VAE and "VAME workflow: training the model and evaluation" for advice on training a VAME model). Within the VAE framework, it is possible to investigate if the model has learned a useful representation of the input data by drawing random samples from the latent space and comparing them to a subset of reconstructions (see Supplementary Section 6 and Methods). After the model is trained on the experimental data ($1.3 \times 10^6$ data points), the encoder embeds the data during inference onto a learned latent space. We then segment the continuous latent space into discrete behavioral motifs using a HMM[31] (Fig. 1a, f)), thereby treating the underlying dynamical system as a discrete-state continuous-time Markov chain (see Methods). See Supplementary Section 1 for more details on model selection, where we also compare the HMM against a k-means algorithm for cluster detection.

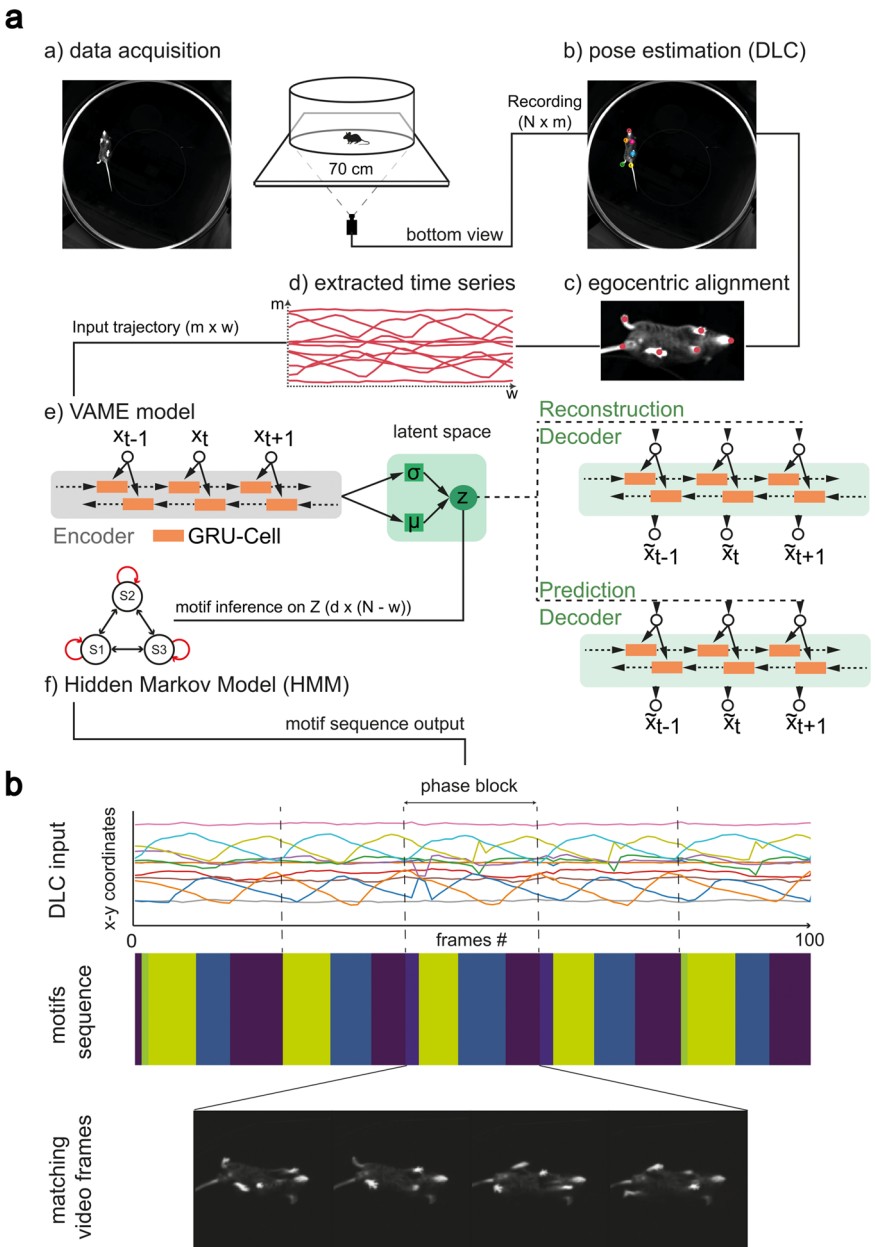

**Fig. 1 VAME: an unsupervised deep learning model for behavior segmentation. a** VAME workflow. Data acquisition via bottom-up camera setup for precise body and limb kinematics. Pose estimation of bottom view (DeepLabCut). Frames are egocentrically aligned and trajectory samples are fed into the recurrent neural network model. The fully trained model resembles a dynamical system from which motifs are inferred via a Hidden-Markov-Model. **b** Example trace of an egocentric aligned DLC time series showing a full walking cycle (phase block). The corresponding motif sequence segmented by VAME has repeated motifs during the phase block cycle. The matching video frames identify the phase block as a full walking cycle performed by the animal.

Figure 1b depicts an example of an egocentrically aligned DLC time series (100 data points). Here, we defined a phase block, which is characterized by a full walking cycle (orange line). The inferred motif sequence aligns to the walking pattern, where the onset of each motif matches to a particular phase of the input signal. The video frames display the corresponding walk cycle within the phase block.

We want to make the reader aware that the Method section provides a full protocol of the VAME workflow with additional information on how to use the method and use it to answer their experimental questions. Here, we discuss the installation of the framework, the necessary preprocessing steps, the training of the model, and the final steps comprising the visualization of the motif time series and the embedding space. Finally, we discuss

common pitfalls and direct the reader to further downstream analysis.

**Identification of behavioral motif structure**. To identify motif structure and to demonstrate the power of our approach, we used $n = 4$ transgenic (tg) mice displaying beta-amyloidosis in cortical and hippocampal regions. These mice harbor human mutations in the APP and presenilin 1 gene (APP/PS1)[27]. We compared these mice to $n = 4$ wildtype (wt) mice housed in identical conditions. For these APP/PS1 mouse line, several behavioral differences have been reported[32]. Among them, motor and coordination deficits[33], changes in anxiety levels[34] and spatial reference memory deficits[35] were most prominently observed. In

the past, batteries of behavioral tests were used to assess possible differences between genotypes since no differences could be detected in open field tests[36]. Hence, this dataset forms an ideal use-case for the purpose of unsupervised behavior quantification to evaluate whether our proposed method can detect those differences.

We determined general locomotor dependent variables to investigate whether the animals show explicit locomotor differences, in particular we investigated speed, traveled distance and time spent in the center (Fig. 2a). The average speed during the experiment was $6.12 \pm 1.36$ cm/s for wt animals and $6.84 \pm 1.57$ cm/s for tg animals with a maximum velocity of $50.61 \pm 12.47$ cm/s for wt and $57.14 \pm 8.91$ cm/s for tg animals. The average time spent in the center (calculated from center crossings) is $9.92 \pm 1.81$ s for wt and $17.14 \pm 7.79$ s for tg animals. Lastly, the average distance traveled was $9187.44 \pm 1266.4$ cm and $9937.07 \pm 1367.08$ cm for tg and wt respectively. No statistically significant differences were found between the groups for all measures. Nevertheless, we see a tendency for the tg group to move at a higher speed and spend more time in the center as already shown by others[36–38]. Moreover, we let human experts classify both groups to identify if the genotype had obvious behavioral differences. The overall classification accuracy was at chance level for all participants ($46.61\% \pm 8.41\%$, Supplementary Fig. 1c). A similar level of behavioral homogeneity between the two animal groups was reported previously[37].

To identify behavioral structure we applied VAME to the entire cohort of animals and inferred the latent representation for each animal. The latent dimension size was set empirically to $z_i \in \mathbb{R}^{30}$, while comparing the difference between input and reconstructed signals as well as the quality of the resulting motif videos (see Methods). We summarize important choices of recording and hyperparameter settings in Table 1 and Supplementary Table 3. Using a HMM on the latent representation, we inferred the same 50 motifs per animal (see Supplementary Section 3 for details) to be able to compare behavioral structure between groups. We then created a hierarchical tree representation $\mathcal{C}$ from the motif structure of the full cohort of animals (Fig. 2b) (see Methods "Motif identification"). By comparing branches of the tree with corresponding motif videos, we identified communities of similar behavioral motifs. We found nine behavioral communities denoted from $a$ to $i$. Each community represents a cluster of movements that can be simplified into actions like e.g., rearing, turning, walking. Motifs within each community can be interpreted as a subset $c \in \mathcal{C}_i$ (with $i = a, \ldots, i$) of these actions. Therefore, communities detected by VAME display a multiscale behavioral representation. All communities were visualized with their respective DLC trace and further described in the Supplementary Section 5. Furthermore, we provide Supplementary Movies 6–14 for all communities.

To find differences between the tg and wt mice we identified up- or downregulated motifs/communities (Fig. 2b). Here, the usage of a motif for the tg group is calculated as a ratio and normalized against the wt group. The *Exploration*, *Turning* and *Stationary* communities are downregulated, while the *Walk to Rear* and *Unsupported Rearing* communities are upregulated. Other communities have certain motifs which are differently used but with no statistically significant group difference (Supplementary Fig. 1).

Most motifs showed a low variance in usage between animals for a given group but there also are motifs where differences between groups are visually apparent (Fig. 2c). To test this observation we compared motifs at the community level between both groups ($n = 8$). A multiple t-Test was used and statistical significance was determined using the Holm–Sidak method with $alpha = 0.05$ (*$P \leq 0.05$, **$P \leq 0.01$). We found a significant differences for five motifs (Supplementary Table 2). *Turning* and

*Stationary* motifs are increased in wt animals while tg animals showed a higher expression of *Unsupported Rearing* and *Walking* motifs, displayed by arrows in Supplementary Table 2. In Fig. 2d we visualized these motifs by taking the start (cyan color) and end (magenta color) frame for a random episode of motif occurrence. White dots are representing the DeepLabCut virtual marker positions over time. Supplementary Fig. 1d shows the corresponding DeepLabCut trajectory of the visualized motif. The motifs are further described in Supplementary Section 2 and displayed in the Supplementary Movies 1–5.

In order to investigate how stable differences during the experiment are, we binned the experiment into six equal blocks (Fig. 2e) and investigated the stability of the five prominent motifs over time. We found that the motif usage is constantly up- or downregulated.

**Motif transitions and behavioral dynamics**. In the VAME framework, discrete representations of animal behavior are organized on different scales, varying from single motifs to communities. On the community level, the temporal structure of behavior can be identified by observing the probability of a community transitioning into another. The resulting transition matrices can be constructed both, on the community level or the motif level (see Methods "Motif identification"). Fig. 3a shows the transition matrices for wt and tg animals ordered by the community structure. It can be seen that both groups share a similar structure of transitions, as expected given the similar open field behavior observed[36–38]. To examine differences in transition probabilities between motifs we created a subtraction matrix $\mathcal{T}_{sub} = \mathcal{T}_{lk}^{WT} - \mathcal{T}_{lk}^{TG}$ that illustrates which transitions are more pronounced in wt animals (Red) or tg animals (Blue). Indeed, we found significant differences in the usage of transitions within communities (Supplementary Figs. 9 and 10). Overall, the most prominent differences in transition probabilities appeared in the *Stationary* community as well as the *Walking* community (Supplementary Fig. 9).

We investigated the *Walking* community in more detail to learn more about the transition differences. When following along the highest transitions on the Markov graph for this community, a cyclic structure appears. Within this cyclic structure, different patterns of walking motifs are more strongly used by both experimental groups (Fig. 3b). To understand this structure, we embedded the encoded latent vectors of the *Walking* community onto a two-dimensional plane via UMAP. We visualized the UMAP (Fig. 3c) for two example animals from the wt and tg group. To determine if this structure is indeed cyclic, we decoded all points back to the original traces containing marker movements. Then, we computed the mean phase for the horizontal hind paw movement using Fourier transformation. We projected the phase angle back onto the embedding of latent vectors and observed that the phase angle follows the curve of the cyclic embedding (red arrow shows phase direction). To parameterize the structure in both animals, we applied k-means clustering. This yielded discrete clusters organized along the cyclic embedding. Such patterns are known to emerge from oscillatory dynamics modeled by RNNs[39,40]. Cyclic representation of walking behavior was recently described in a *Drosophila* locomotion study[41] and here we confirm the existence of this representation also in rodent locomotion. We explore the possibility to detect locomotion subpatterns from this embedded dynamics in the Supplementary Section 7.

**Quantitative comparison of VAME with MotionMapper and AR-HMM**. Several approaches for behavioral quantification exist, which all lead to valuable neuroethological data and provide

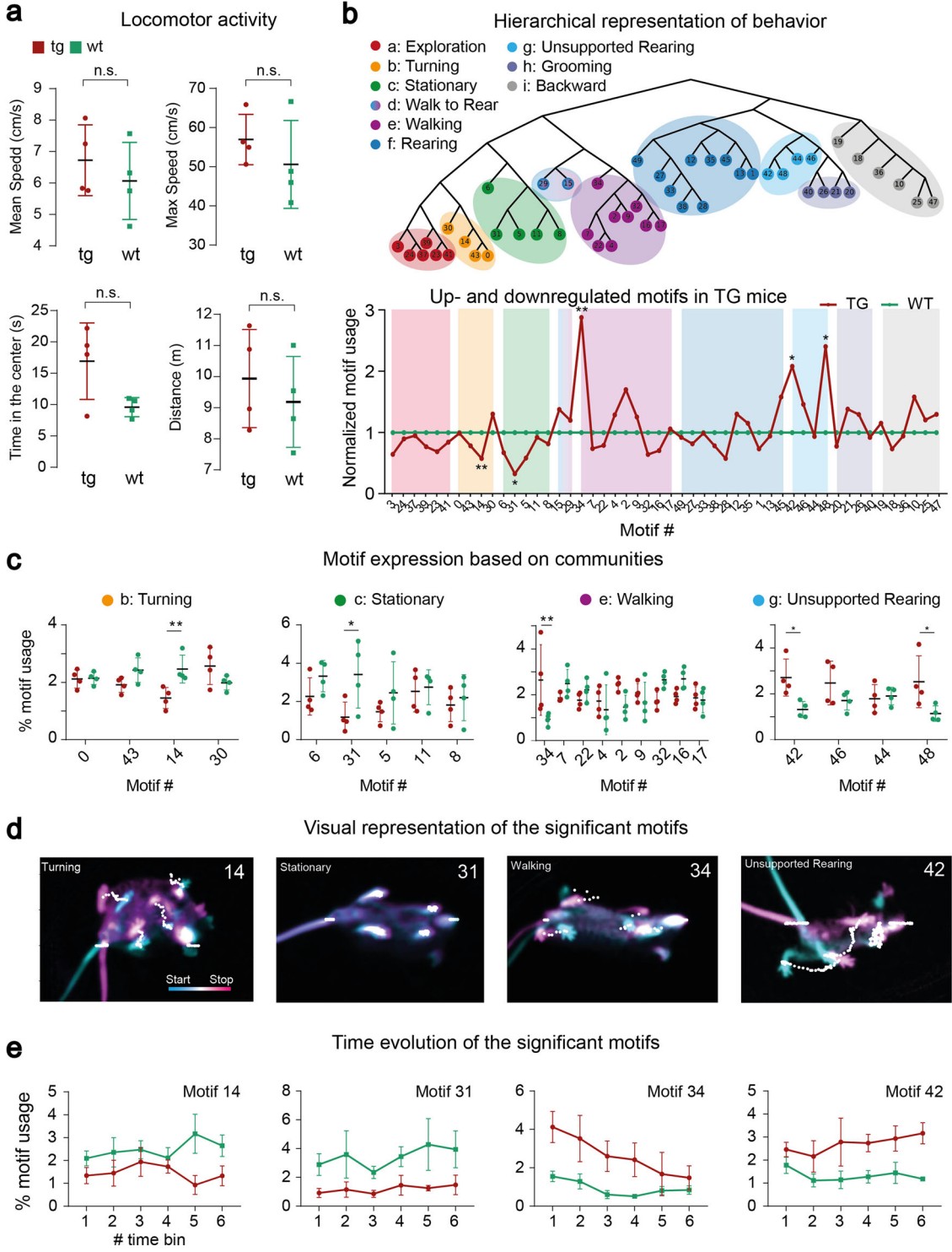

**Fig. 2 Behavioral quantification with VAME and hierarchical community clustering. a** Locomotor activity of transgenic (tg, $n = 4$) and wildtype (wt, $n = 4$) animals. **b** Hierarchical representation of behavioral motifs. Color grouping on the tree are depicting communities of motifs which belong to the same observable category of behavior. A depiction of up- and downregulation of motifs and communities in tg animals (red line) compared to wt animals (green line) and ordered by communities is shown below. **c** Quantification of motif usage in percent (%) ordered by communities. Differences between the tg and wt phenotype are in community b, c, e and g. **d** Visual representation of significantly changes motifs. The start frame is colored in cyan and the end frame is colored in magenta. White dots represent the DLC virtual marker points. **e** Time-dependent modulation of significantly changed behavioral motifs for both phenotypes binned into six blocks. Error bars represent standard deviation.

important means for understanding the neural correlates of behavior in different model organisms[10]. Since VAME makes use of the variational autoencoder framework, the approach differs substantially from others. We perfomed a qualitative and

quantitative comparison with two existing approaches (see also Supplementary Section 8).

To validate the models (VAME, AR-HMM, MotionMapper) we created a manually labaled dataset that was annotated by

**Table 1 Important parameters of the** `config.yaml` **file.**

| Parameter | Default value (*type*) | Description |
|---|---|---|
| model_name | VAME (*string*) | Name of the model |
| n_cluster | 30 (*int*) | Number of motifs |
| pose_confidence | 0.9 (*float*) | Minimum accuracy from pose estimation that will be accepted. |
| project_path | working directory path (*string*) | The path to the working directory. It can be adapted if the project is moved on the file system. |
| video_sets | - video-1 (*string*)<br>- video-2<br>- ... | Set of video files that are used as input to VAME. |
| egocentric_data | False/True (*boolean*) | Specifies if the data is egocentric or needs to be aligned. |
| num_features | 12 (*int*) | The total number of markers (both *x* and *y*) in the input CSV file. |
| time_window | 30 (*int*) | The size of the time window that moves through the data and is passed to the model. |
| zdims | 30 (*int*) | The number of latent dimensions that the model embeds the data to. See chapter XX for further discussion. |

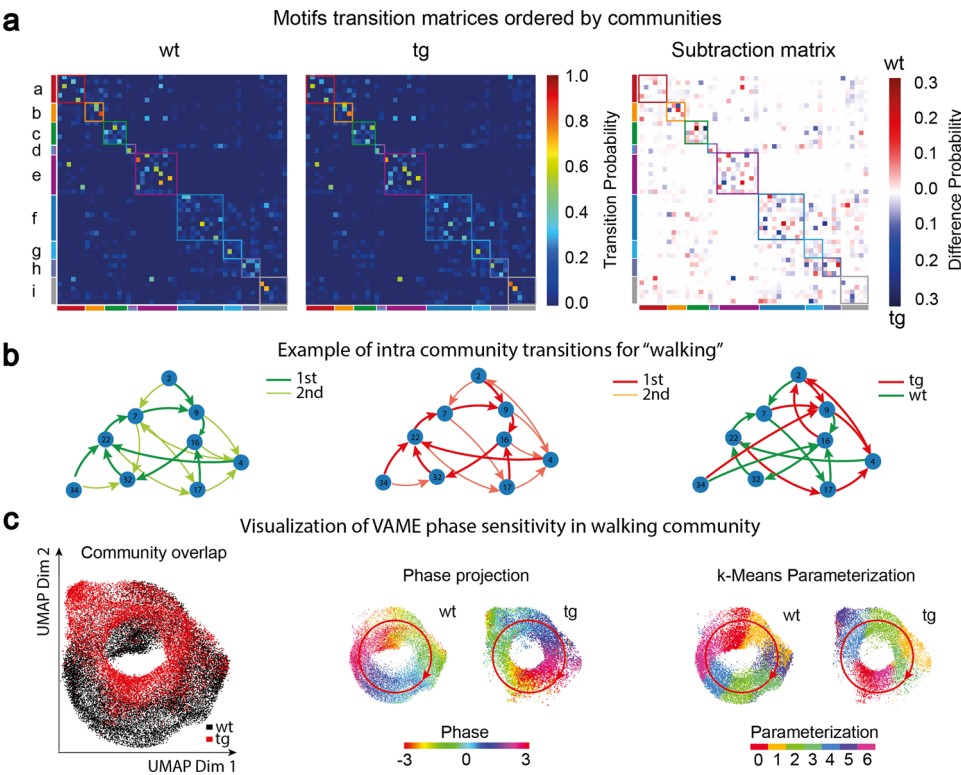

**Fig. 3 Identification of transition structure and locomotion patterns. a** Transition probability matrices ordered by communities for the wt and tg group and the corresponding difference plot of both matrices. Squares along the diagonal indicate the grouping into communities. **b** Example of an intra-community transition graph for the walking community. The first two graphs are showing the two highest transitions for both groups and the third graph shows the highest transition difference. **c** Joint UMAP embedding of points belonging to the walking community in a wt (19.783 points, black) and a tg (13.264 points, red) mouse reveals a circular structure. The projection of the mean phase angle of the horizontal hind paw movement onto the embedding displays the cyclic phase space of the walking movement in both animals. Parametrization of both point clouds with k-means shows blocks organized around the cyclic structure. Red arrow indicates the phase direction.

three human experts with training in behavioral neuroscience. A video of a freely moving wt animal consisting of 20.000 frames (≈6 min length) was annotated with five behavioral labels (Walk, Pause, Groom, Rear, Exploratory behavior) (Fig. 4a, see Methods "Manually assigned labels and scoring"). When quantifying agreement between individual experts, we observed that 71.93% of the video frames were labeled equally by all three. The remaining 13.61% of frames were labeled unequally by two experts and 14.47% were labeled unequally by all three experts (Fig. 4b). This implicated that behavior showed a considerably high observer variability and is not trivially assignable to discrete labels[6,10].

We trained all models on the full dataset and validated how they overlapped with the manual annotation (Fig. 4c). Here, the blue columns represent the annotater agreement per motif of the given model. The red columns represent a given models purity based on discovered motifs and expert labels. We indicated an annotator agreement of over 90% with a given model by a black box in both (blue and red) columns. VAME had 16 motif overlapping with a high annotator agreement (>90%) (Fig. 4c, black boxes). For MotionMapper, we identified 11 motifs which had a high annotator agreement and for the AR-HMM we identified 5 motifs with a high annotator agreement. This suggests that VAME is able to detect human identifiable labels in a more

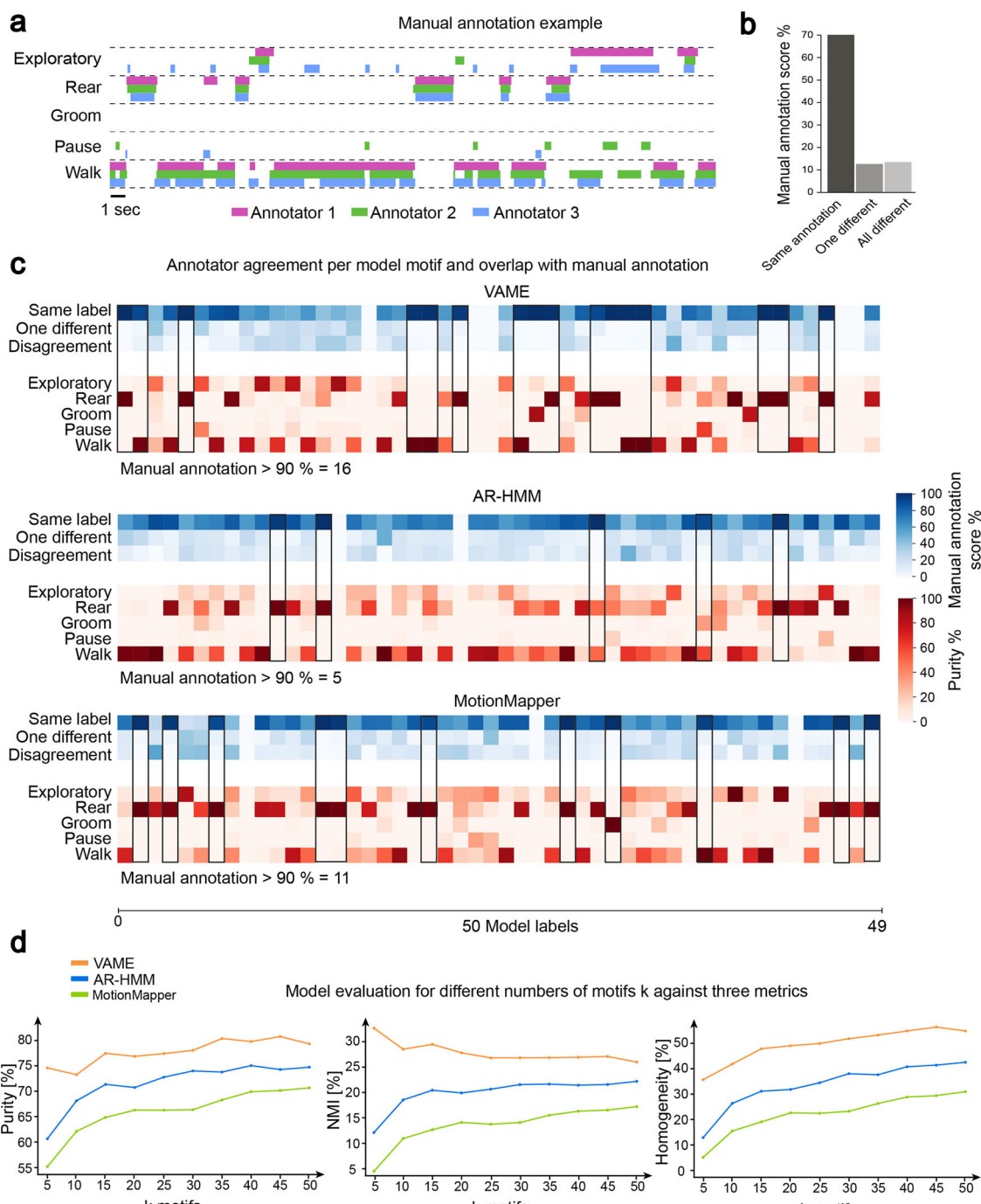

**Fig. 4 Annotated dataset and model comparison based on annotator agreement. a** Overlap of manually assigned labels by three experts. **b** Disagreement in manual annotation. **c** Confusion matrices showing the annotator variability (blue) and the agreement between 50 model (VAME, AR-HMM, MotionMapper) motifs and 5 manually annotated labels (red). Empty columns exist when the specific motif did not appear in the annotated benchmark data. **d** Model evaluation on three metrics (Purity, NMI, Homogeneity) as a function of number of motifs $k$.

concise way than both other models. Lastly, it can be seen that columns are sometimes empty. This is the case as the models were trained on the full dataset but the inference was only done on the annotated dataset which was a smaller subset (0.8% of the full dataset herein). VAME has five empty columns while the AR-HMM and MotionMapper have two and three, respectively. This might indicate that VAME is more selective about motifs and not all motifs are present in the smaller benchmark dataset.

To further investigate the overlap of each model with the benchmark dataset we quantified Purity, Normalized Mutual

Information (NMI) and Homogeneity (see Methods "Manually assigned labels and scoring"). Applying all three measures we found that VAME had the highest score for each measure (Purity: 80.65%, NMI: 28.61%, Homogeneity: 54.89%) (Supplementary Table 4), when applied to a motif number of $k = 50$. In Fig. 4d, we further showed that VAME achieves the best scores on all three metrics when measured as a function of motif number $k$. Interestingly, the performance of VAME stays stable even for small motif numbers compared to the AR-HMM and Motion-Mapper. Additionally, we passed the original pose data also to a

standard gaussian emission HMM and applied all three metrics to the outcome to rule out that the performance of VAME is only determined by the downstream HMM. Here, we found that the HMM performance is similar to MotionMapper and significant lower than our approach (see also Supplementary Table 5 for different sizes of $k$.) In Fig. 4c, we furthermore compared agreement overlap of the annotator with the model motifs. Here, we found that VAME shows the highest overlap followed by MotionMapper and AR-HMM. Moreover, in Supplementary Section 9 we investigate the spatiotemporal embedding of the VAME latent space by downprojecting it via Unifold Manifold Approximation (UMAP) and compare it to the MotionMapper t-SNE space (Supplementary Fig. 6).

## Discussion
In this manuscript, we introduce an unsupervised deep learning method called Variational Animal Motion Embedding (VAME) to discover spatiotemporal structure of behavioral signals obtained from pose estimation tools. It combines the VAE framework[17] with autoregressive models (biRNNs) and creates a probabilistic mapping between the data distribution and a latent space. This approach enables the model to approximate the distribution and to encode factors that contain sufficient information to separate distinct behavioral motifs. A structural element of this model that differentiates it from other approaches is that it uses an additional biRNN decoder that learns to anticipate the structure of the subsequent trajectory and regularizes the latent space. This addition forces the encoder to learn improved dynamical features from the behavioral time series. We want to stress that we use the term *behavioral dynamcis* freely within this work as a synonym for time-dependent analysis of body part movements. The biRNN model within VAME is effectively carrying out a fit of the gated recurrent unit equations to the pose estimation signal, which in a data-driven way describe the motion of the DLC markers via difference equations.

VAME addresses a pressing need for behavior quantification in neuroscience, because current methods still insufficiently capture the complete spatiotemporal dynamics of behavior[10], and limit our understanding of its causal relationship with brain activity. The field uses a rapidly developing repertoire of experimental approaches to investigate neural correlates of behavior[42–44]. Monitoring neural activity in freely behaving animals with imaging, electrophysiological tools and cell-type specific interrogation methods are state-of-the-art. In addition, new and traditional transgenic animal models allow for a deep investigation of molecular pathways in health and disease. All these approaches require deep, reliable and complete dissection of behavior. In this manuscript, we used a traditional transgenic model of Alzheimer's disease, the APP/PS-1 model, to demonstrate discriminatory power of our approach in a mouse model system, which shows clear behavioral deficits in specific tasks, but no reported differences in open-field observation[36,38]. Our approach, however, is not limited to any specific species or behavioral task as long as continuous video-monitoring can be provided[16,45].

Our results demonstrate that VAME is capable of detecting differences between a tg and wt cohort of mice, while no differences were found by the human observer. In our use-case, we did not aim at investigating behavioral deficits in the domain of learning and memory with relation to Alzheimer's disease. However, even in this small sample size, VAME identified a higher motif usage of the *Unsupported Rearing* community as well as a lower motif usage in the *Stationary* community that could be related to deficits of spatial orientation and environmental habituation[34,35]. Interestingly, the motif usage within groups showed a low variance, which points towards the robustness of the method to detect a common and stereotyped behavioral structure.

We found that VAME is particularly suited to learn motif sequences from pose estimation signals. The main reason for this is that VAME is equipped with a particularly high sensitivity to the phase of the signals due to its biRNN encoder and decoder. Here, we showed this by plotting the phase angles onto a two-dimensional UMAP projection for the walking community. In this way, we could uncover a circularly organized point cloud, which exactly captured the natural cycle of limb movement[41]. Thus, VAME may be particularly useful in the detection of reoccurring locomotion patterns. While we so far have shown this only in one community, this advantage could be further exploited to identify differences in other movement types.

While VAME is effective in learning motif sequences from pose estimation signals, we also investigated the hierarchical structure of the resulting motif sequence by creating a tree representation. Within the VAME framework, motifs are sub-patterns of macro behaviors organized on a Markovian graph (see ref. [45]), Figure 3 (left)). By considering transition and usage properties on this graph, we can identify different types of e.g locomotion as shown in Supplementary Fig. 7. Hence, the tree representation transforms the motif sequence into broader, human readable categories like *Walking* or *Rearing*. This feature, though, is not unique to VAME and our approach to transform sub-patterns (or motifs) of behavior into a tree representation could be applied to any other supervised or unsupervised method.

VAME models the spatiotemporal information to segment behavior comparable to MoSeq and MotionMapper[7,11]. To incorporate spatiotemporal information, MoSeq applies an AR-HMM to infer hidden states from a series of transformed depth images. On the other hand, MotionMapper, which was initially implemented for fruit flies, relies on t-SNE[46] embeddings of wavelet transformations from high-speed camera images. In this method, regions with high densities are assumed to contain stereotypical behaviors. Since the spectral energy of a signal is the key input feature, low frequency movements, which are more prominent in mice than in flies, limits capturing the full behavioral repertoire. In contrast to MotionMapper, MoSeq was first applied in freely moving rodents. This allowed the detection sub-second behavioral structure but the AR-HMM resulted in a multitude of short and fast switching motifs, which can lead to uncertainty in animal action classification. These three methods have all been successfully applied to capture the behavioral dynamics in different animal models and experimental settings. To compare their performances, we trained all models on our data and investigated their motif sequence distribution on a benchmark dataset. Each model learned a consistent motif structure, nevertheless, VAME obtained the highest scores for all three metrics (Purity, NMI, Homogeneity). This result could be due to a better embedding of the spatiotemporal information and higher phase sensitivity of our model that is not as strongly present in the others. Recently, an independent group of scientists has published comparative data obtained from testing VAME to other approaches on a benchmark dataset for a hand-reaching task[16]. Their findings fully support our own observation of higher VAME performance levels against other approaches (AR-HMM, Behavenet[7,47]) in the following three criteria: Accuracy, NMI and Adjusted Rand Index. Indeed, the combination of the video representation model used in this study with VAME achieved the best results.

In the light of our comparison between methods, we want to highlight that the future of computational ethology needs improved benchmarks and datasets for single and multi-animal behavior. Comparing methods on one particular dataset can shed

some light on their performances but certain methods are better suited for certain kinds of data. With the rising amount of tools for computational ethology this becomes a pressing need and we want to trigger the development of better benchmarks and metrics that do not only rely on motif structure but also consider representations in lower dimensional space. Here, we showed that VAME is indeed capable of providing good solutions to both (Supplementary Figs. 8 and 9) but we have to acknowledge that this was done in one particular setup and dataset. Future benchmarks need to evaluate these tools on much broader scenarios to come to a conclusion about when to apply a given method.

An important aspect of VAME are the choices of its hyperparameters, which we summarized in Supplementary Table 3. Here, the number of latent dimensions is one of the central parameter. The embedded dimension controls the amount of information the model can exploit. Based on the information bottleneck theory[48], this number should be chosen as small as possible to extract the most essential information from the data. However, this is coupled with multiple factors. One of the principal factors for VAME is the choice of time window $w$ and number of marker coordinates $m$ as this is the information VAME is condensing into a vector representation $z_i$. Higher numbers of $w$ or $m$ (or both) can result either into expanding the latent dimensions or into preprocessing the data through a top layer neural network (or other techniques like principal component analysis). Users of VAME need to adjust this number to their needs and should be aware that this number significantly affects the outcome of their VAME model. Having a benchmark dataset and investigating the reconstruction score can help to identify an appropriate number.

Furthermore, choosing the number of appropriate motifs is another question, which is hard to generalize, as every experiment and/or animal used will have their unique set of behaviors and hence number of motifs. In this work, we considered only motifs that have a higher than 1% usage after sampling 100 motifs from our embedding space and re-running the motif segmentation with this number. In general, however, it would be of special interest to identify motifs that are present in one group/animal but not the other. This would show a complete different set of behavior and mark highly significant differences between them. Our data is very homogeneous in terms of behavior and the two populations do not differ drastically in their executed behavior. Hence, such motifs are not likely to appear in this work. However, we believe that VAME is capable of finding these "out-of-distribution" motifs when they exists based on the variational autoencoding framework. In general, for every behavioral quantification method there is a trade-off of how general the motif distribution should be versus how precise individual behavior is measured. If the goal is to identify very specialized individual behavior it would be possible to parameterize both populations or all animals individually. The problem lies in relating motifs with each other between populations/animals as the motif mapping would change per parameterization. Users of these tools should keep this in mind and this questions needs to be addressed by future work.

Applying a trained VAME model to other unseen animal datasets can be beneficial in terms of reducing computational costs and identifying similar motif distributions. A caveat, however, lies in the fact that the unseen data must follow approximately the same data distribution as the training and test set. This is typically the case if several animal recordings have been captured from the same cohort, under the same circumstances, camera setup, etc. However, in general, we recommend users to include the data to be segmented into motifs also into the training dataset, as detailed within the Methods.

While VAME yielded higher performance scores when compared to other unsupervised approaches, supervised approaches may be better suited in experiments in which obtaining the full behavioral repertoire is not required. Supervised approaches like SimBa, MARS, or DeepEthogram[2–4] all allow the labeling of episodes of interest. Lastly, B-SOiD[12], a recently developed unsupervised method that does not require a deep learning model, allows for a fast identification of similar frames and trains a classifier to identify these clusters rapidly in new data points. This approach, however, does not use a trajectory sample of the behavior and projects framewise into a UMAP representation. The temporal information comes mainly from a velocity feature signal. These aspects should be considered when choosing an optimal behavioral quantification method for a specific task and species. As VAME learns spatiotemporal information, it may be particularly useful to uncover behavioral dynamics in a lower dimensional latent space. Moreover, VAME also has the potential to train a classifier on the latent vector information to quickly assign VAME motifs to new data points.

A promising application of VAME could be the combination with three dimensional pose information[49–51], which can be easily incorporated into the VAME model. This will likely lead to a better resolution of behavioral motifs, as most behaviors are expressed in three dimensions. When aiming at quantification of behavioral information with even higher dimensionality, the VAME model allows for a straightforward integration of parameters such as cellular calcium responses, neurotransmitter dynamics or other physiological features.

Taken together, we believe that VAME is an extremely useful method for unsupervised behavior segmentation, that can be easily applied by other scientists and strongly facilitate the investigation of causal relationships between brain activity and visible naturalistic behavior. VAME can identify motif structure from pose estimation signals with a high degree of generalization between animals and experimental setups. The framework is open-source and easily accessible without expert knowledge in programming and machine learning, and thereby open to a wide audience of neurobiologists and other scientists. We anticipate that it will stimulate the development of further machine learning models[16,52,53] and will trigger the development of robust metrics and benchmark datasets for the computational ethology community.

## Methods

**VAME workflow: overview and installation**. In the former sections we presented the VAME model, demonstrated its abilities on a use-case of APP/PS1 animal to identify motif structure as well as embedding a dynamical space in which the phase is well preserved. We provide the complete framework as an open-source package on our GitHub website (https://github.com/LINCellularNeuroscience/VAME). In the following section we aim to guide users through the installation process and workflow, from setting up a VAME project, preparing the training data and fit the model to common pitfalls and downstream analysis.

VAME is a general time series quantification method and while we used in our exemplary data pose tracking input from *DeepLabCut*, VAME works also with other pose estimation tools like SLEAP, DeepPoseKit or B-KinD[14,15,54]. In principle, other kinds of data such as a principal component time series of the video data or other sensory signals can be fed into the model. Throughout this protocol we will use the demonstration data that are available on the VAME GitHub page, which is a video of a freely moving mouse in an open-field arena (*video-1.mp4*) and the corresponding DLC file containing the coordinates of the virtual markers (*video-1.csv*). The dataset contains 29,967 frames. Note that it is possible to train a working VAME model with as little data as this to achieve good results in terms of motifs and latent space dynamics.

We recommend to install VAME within an Anaconda (https://www.anaconda.com/products/distribution) virtual environment, for which we provide installation files on the GitHub project website. Upon fetching the most up-to-date codebase from GitHub, a new environment containing all necessary dependencies can be created with the command `conda env create -f VAME.yaml` in the Anaconda prompt. Make sure you navigated to the directory where you fetched the GitHub repository of VAME, e.g., `C:\MyUser\GitHub\VAME\`. To install

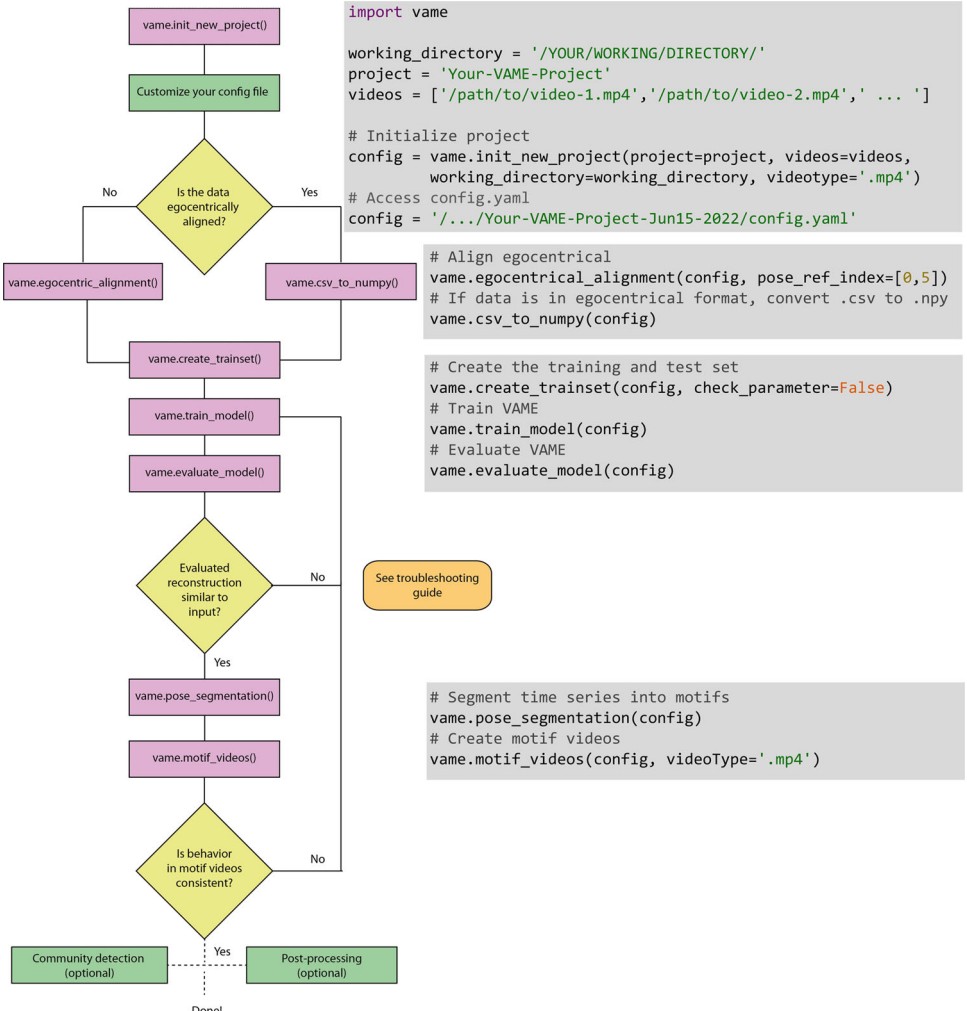

**Fig. 5 Workflow figure for VAME and the corresponding code functions.** The figure shows the complete life cycle of a VAME project. The main steps are the project initialization, the transformation of the data from the pose .csv file to a Python .npy file, the creation of a training and test dataset, to training and evaluating the model, and lastly to segment the pose data into behavioral motifs. Afterwards, user can invest behavioral motifs by creating videos from these episodes.

VAME activate the new environment in the Anaconda prompt via `conda activate VAME` and type `python setup.py install` in the command window, which concludes with the full installation of the package. Finally, users should consider to additionally install the Anaconda extension Spyder (`conda install spyder=5`), which is an integrated development environment (IDE) and is useful to work through our provided *demo.py* code as well as debugging and adding analysis.

VAME utilizes specialized hardware, i.e., graphics processor units (GPU). In this work, we were using a single Nvidia GTX 1080 Ti GPU to train our network. Users should either have local access to a GPU or use a cloud computing provider like Google Colab (https://colab.research.google.com/). To use Google Colab with VAME, one has to move the fetched GitHub repository onto a Google Drive and mount the path within Google Colab. Moreover, VAME can be trained on a central processing unit (CPU), but this leads to longer training and inference time. For an optimal introduction into the setup and hardware for modern behavioral experiments we recommend the recent review paper about Open-Source tools for behavioral video analysis[55].

**VAME workflow: initializing a new VAME project**. The VAME workflow starts by initializing a new project with the function `vame.init_new_project()`. It takes in four arguments; the project name, a path to the directory of the animal videos, a path that specifies the working directory where the project folder will be created, and a parameter that specifies if the used videos are `.mp4` or `.avi` (Fig. 5, first gray box). The user needs to spell out the full path to a video such as `/directory/to/your/video-1.mp4`, otherwise the `config.yaml` file is not correctly initialized. This will create a folder with the project name and the creation date e.g., `Your-VAME-Project-Jun15-2022`. Within this folder four sub-folders will be created (`data`, `model`, `results` and `videos`) and a

`config.yaml` file, see Fig. 6 for reference. Note that the *video-1.csv*, which contains the DLC pose estimation output, needs to be put manually into the `pose_estimation` folder.

Once the VAME project folder structure is correctly setup, the user can continue to check their configuration before starting to prepare the data and training the VAME model. The `config.yaml` file is essential for your VAME project as you can set all the necessary parameter in here. Certain parameters have to be set manually for the used dataset, while others are default parameter for the neural network architecture used by VAME. These parameters should be only changed if the user has the sufficient understanding of PyTorch and the contained models. The parameters that are necessary to set by the user are summarized in Table 1.

**VAME workflow: egocentric alignment and pre-processing**. Typically, data from pose estimation tools provide coordinates of tracked body parts as 2D coordinates that have been extracted from the input video. Using DLC, this data can be stored in a *.csv* file. If the data is from a freely moving animal (open-field arena, operant conditioning chamber, etc.), the data needs to be egocentrically aligned first. For this purpose, VAME provides the function `vame.egocentric_alignment()` (Fig. 5), which aligns the animal along its principal body axis (here spine). If, on the other hand, the data comes from a head-fixed setup (or similar), where the global virtual marker coordinates align with the egocentrical coordinates of the animal, one can simply transform the *.csv* file to a VAME readable *numpy* file with `vame.csv_to_numpy()`. Its argument is the `config.yaml` path (Fig. 5).

The `vame.egocentric_alignment()` function takes in four arguments: the path to the config file as well as the indices of the virtual markers (according to the input *.csv* obtained from the pose estimation) that are used as reference points

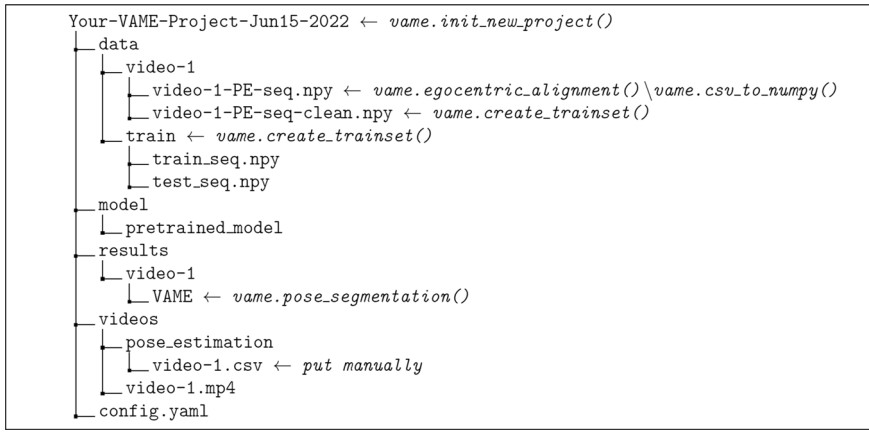

**Fig. 6 VAME project folder structue for one video (*video-1.mp4*).** By initializing a VAME project a project folder will be created within the working directory. This folder contains the raw data, the configuration file (config.yaml), the trained VAME model(s), and the results (segmentation, latent vectors, motif videos).

for the transformation to egocentric coordinates (`pose_ref_index`). As for our example, the data obtained from the bottom-up recordings of a laboratory mouse (see Fig. 1a as example), we might use the tuple [0, 5], as the 0th marker refers to the snout tip and the 5th marker refers to the tail root of the mouse (see *video-1.csv* for reference). The egocentric alignment function then transforms all other marker points to the linear axis defined as the vector between the two reference points. The other two arguments are the `crop_size` parameter, which defines the maximum pixel area in the video that contains the complete body (and all virtual marker) of the animal. This parameter is only necessary to check if the alignment fails or works poorly. The fourth parameter to the function is a boolean parameter called `use_video`. If this parameter is set to `True`, a video output of the egocentric alignment is created and the user can visually inspect if the egocentrical alignment worked. Note, however, that in this setting the function takes significantly longer to run. It is recommended to also check the resulting time series for a signal that is not skewed and cut-off at some extreme values.

Using two appropiate anchor points for the egocentrical alignment is crucial to transform the animal from its allocentric virtual marker coordinates to its own egocentric coordinate system. However, these points can be sometimes occluded by e.g., the animals body. If this happens, the pose estimation will output a low accuracy of the respective keypoint and the `vame.egocentric_alignment()` function will set the value of this keypoint to NaN ("Not a Number"), which will be later interpolated when creating the training dataset.

After extracting the virtual marker coordinates and bringing them into the right format, we can now create the training data for VAME and apply some preprocessing if necessary. For this, the user can use the `vame.create_trainset()` function (Fig. 5, third gray box). For this function to work properly it is important to set the configuration file parameter `egocentric_data` correctly. If set to `True`, the `vame.create_trainset()` function assumes that the data is egocentric by design and does not define anchor points, which were used to align data that is not egocentrical a priori. Now, the function takes in two arguments; the `config.yaml` path and a boolean option `check_parameter` and applies some filtering to the data, which can be controlled by the user. Specifically, VAME implements a Savitzky-Golay filter to smooth the time series as well as some thresholding based on the interquartile range (IQR) to robustly cut out outliers within the signal. To set the IQR value and filter settings, the config.yaml provides the following parameter: `robust`, `iqr_factor`, `savgol_filter`, `savgol_length` and `savgol_order`. The first parameter will eliminate outliers in the signal if set to `True` by multiplying the IQR value with the set `iqr_factor`. If a value is higher than this it will be set to NaN. Equally, if a value is smaller than the negative of this result, it will be set to NaN. Afterwards, the NaNs are linearly interpolated. The second parameter defines if a Savitzky-Golay is being used on the data while the last two specify the length and order of that filter respectively. It is advised to carefully check how these parameter act on the original signal, hence the function provides the `check_parameter` argument. If set to `True`, the function runs only on parts of the data and displays the original signal and the filtered signal (Supplementary Fig. 11). Here, the user can check the IQR threshold and compare the filtering to the original data.

**VAME workflow: training the model and evaluation**. In the previous step, we made sure that the input data of the pose tracking is correctly aligned and pre-processed. This data is stored in the folder `\Your-VAME-Project-Jun15-2022\data\video-1\video-1-PE-seq-clean.npy\`. A new folder has been created called `train` (Fig. 6), which stores the data for training the model and hold out data for testing the model. By varying the `test_fraction` parameter in your *config.yaml*, the user can decide how much of their data is used for testing. The default value is 0.1, which relates to 10% of the data.

To train your VAME model, the user can call the function `vame.train_model()`. The input arguments to this function are the *config.yaml* path. Within your configuration file, the most crucial parameter to train a VAME model are the `time_window` and `zdims` parameter (see Table 1). We recommend to set the first variabe, which defines the input trajectory length, to at least 20 datapoints for the RNN to work properly. The reasoning here is that we want to learn an embedding from the kinematic trajectory of the moving animal. Higher values of this can also benefit the model and users are adviced to train multiple models with different settings to identify an optimal solution for their data. The other parameter, `zdims`, ensures that the model embeds a lower dimensional representation of the input trajectory. To identify a good setting of this parameter, users could train their model with different settings and evaluate when the reconstruction and prediction mean-squared-error losses plateau. For the demonstration data we found that the plateau startet at around 12 latent dimensions (Supplementary Fig. 13).

The configuration file further provides parameter to set the batch size, learning rate and other parameter of the network. Users without experience in training deep neural networks are adviced to keep these parameters as default. For more explanation of the different configuration file parameter, please consult our GitHub wiki page https://github.com/LINCellularNeuroscience/VAME/wiki/2.-VAME-config.yaml. The training process ends after a convergence criteria has been met, which is defined by either the maximum number of epochs or if the loss on the test set has not improved over a certain amount of epochs (default 50).

Once the model finished training, the user can inspect the model with the function `vame.evaluate_model()`. This function takes the same arguments as the `vame.train_model()` function. The trained model will be evaluated on the test set and the output of the evaluation are two plots that show the reconstruction and prediction capabilities as well as the loss curves (Supplementary Fig. 12). Here, the upper part shows a model with good reconstruction capabilities of the signal and which can predict the 15 subsequent time steps very well. The lower part shows a model that has poor reconstruction and prediction capabilities. In this case, one has the option to increase the amount of training data, increase the bottleneck by setting `zdims` to a higher number or inspect the training data to make sure that there are no obvious outliers (Refer to Supplementary Fig. 11 and the troubleshooting section below for more advice).

**VAME workflow: model inference**. After the successful training of a VAME model it can be used to infer the motif structure within the dataset i.e., segmenting the behavioral time series into discrete units. This is done with the function `vame.-pose_segmentation()` (Fig. 5, fourth gray box). The critical parameter to set is `n_cluster` in the configuration file, which defines the number of discrete units (or motifs). While this number is usually not known a priori, it can be useful to run the analysis described in the Supplementary Section 3. Here, we run `vame.pose_-segmentation()` function first with 100 discrete states and defined a threshold at the 1% motif usage mark. This led to 50 motifs that could be present in our data. Next, we re-run the function with this number to get the final result. Users, however, should always cross-check the quality and validity of the motif videos after the segmentation with `vame.motif_videos()`.

Finally, it is important to note that the inferred motifs time series is shorter than the original input time series. This stems from the fact that we use a `time_window` (see section above) that represents the trajectory length, which is embedded by the RNN encoder into the latent space. By using a `time_window = 30`, the first motif label starts at the 15th frame and the last label ends at the $(N − 15)th$ frame, with $N$ being the full length of a behavioral video.

**VAME workflow: visualization and post-processing**. As already mentioned in the subsection before, the discrete motif time series can be visualized as single motif instance videos with the function `vame.motif_videos()` (Fig. 5, fourth gray box). The function samples several sequences of each motif into a video. Here, the configuration file parameter `length_of_motif_video` controls the length of each motif video. By visualizing motifs as videos it is possible to check if they are consistent. Note that due to noise or other unregularities from the input data, some misclassifications can occur. Moreover, the user can also visualize the embedded latent space with the function `vame.visualization()`. This function generates an Uniform Manifold Approximation and Projection (UMAP) embedding from the original latent space (in this example 12 dimensions) into a 2 dimensional space.

Finally, it is worth exploring some of the generative capabilities of VAME to learn more about the models representation of the motif or general latent structure (see also Supplementary Section 6 for more information). Here, we provide the function `vame.generative_model()` that takes three input arguments, the configuration file, a string called `mode` and an integer called `motif_num`. The argument `mode` can be set to three options: *reconstruction*, *sampling* and *motifs*. Here, the first two are generating random trajectory samples based on either reconstructing orginal samples or generating new samples from the distribution (length depends on your configuration file parameter `time_window`). If mode is set to *motifs* and `motif_num` to *None*, the function will generate ten new samples for each motif. If `motif_num` is set to an integer, it will generate new samples for a specific motif defined by its integer number.

**VAME workflow: pitfalls and downstream analysis**. Finally, it is important to highlight some common pitfalls when using VAME, especially for new users. This is by no means an exhaustive list but points to general questions VAME users encountered:

- *Virtual marker identity switches:* VAME can only be as good as the provided input signal. Inconsistent motifs can occur fast when, for example, the pose tracking method is not trained well enough and identity switches between keypoints occur. Identity switching of virtual marker can have a strong effect on the learned embedding space of VAME and reduces the models capability to identify robust dynamics and motifs.

- *Strong transients or noise:* Another important aspect is to make sure to check the virtual marker signal for big transients or strong noise before training the model. Otherwise, this could lead to the RNN ignoring the actual behavioral signal. It is adviced to crop the video to the actual behavior/experiment.

- *Check the training dataset:* Another common issue (related to the first two) is the quality of the training data. VAME provides options to preprocess the data within the `vame.create_trainset()` function, which can be control via the configuration file. By additionally setting the argument `check_parameter` of the function to `True`, the user can inspect their data. The Supplementary Fig. 11 provides some intuition on what users should look for.

- *Perspective (top down vs bottom up):* The presented data in this manuscript comes from a bottom-up perspective. This perspective reveals a lot of kinematical details of the freely moving animal, especially the limb movements, which VAME can extract very well into a dynamical space (see Fig. 3c). Moreover, with this perspective it is easy to identify differences in locomotion. In contrast, using a top-down perspective, the virtual pose marker capture less of the limb kinematics. In this case, it is advised to add additional information to the model like velocity and acceleration for the center-of-mass or per marker to improve the embedding.

- *Perspective (scale differences):* If animals are recorded in different setups or the camera angle/distance might slightly differ from animal to animal it is important to take into account these differences in scale. We provide in our `vame.create_trainset()` function z-scoring for the complete dataset. However, user should check that all time series are following the same statistics. Different scaling will most likely result into different embedding of the model. This is also important if different group of animals are used e.g., male and female, where one group is bigger in size. We advice to make sure that the resulting time series for training and embedding do not differ in scale, otherwise the resulting motif structure might differ between animals.

- *Hyperparameters:* We discussed the crucial hyperparameters like trajectory length and latent dimension size within this section. It is important that users keep in mind these parameter and adjust them to their needs or train multiple model to identify an optimal hyperparameter setting.

When finally having successfully trained a VAME model, users have the extracted motif time series of their data and the latent space vectors within the results folder. Here, we want to highlight some downstream analysis that users can engage with. The obvious starting point is using the extracted motif time series. The time series can be used to detect differences in usage between two groups or on an individual level. Furthermore, it is possible to create the hierarchical representation of the time series to identify the grouping of each motif. By spanning a graph network, as done in our prior work[45], users can study different connections

between motifs and most likely paths and transitions during experiments. Our GitHub code provides some functionality for this but custom analysis scripts and extensions will be needed. If the goal is to correlate the motif time series of VAME to neural activity, we recommend reading our prior work, where we use an information theoretical approach to compare the similarity of lower dimensional representations of neural activity during different motifs and communities.

Another possibility provided by the VAME output is the study of the lower dimensional behavioral latent space. We showed that it is possible to map communities into a 2D representation from their original VAME space and to study dynamics of e.g., locomotion. This can be extended to any community or motifs of interest. Lastly, these latent vectors can also be used to train classifiers on top of VAME for the fast detection of behavioral motifs.

**Animals**. For all experiments we used 12 month old male transgenic and non-transgenic APPSwe/PS1dE9 (APP/PS1) mice[56] on a C57BL/6J background (Jackson Laboratory). Mice were group housed under standard laboratory conditions with a 12-h light-dark cycle with food and water ad libitum. All experimental procedures were performed in accordance with institutional animal welfare guidelines and were approved by the state government of North Rhine-Westphalia, Germany.

**Experimental setup, data acquisition and preprocessing**. In the open field exploration experiment mice were placed in the center of an circular area (transparent plexiglas floor with diameter of 50 cm surrounded by a transparent plexiglas wall with height of 50 cm) and have been left to habituate for a duration of 25 min. Afterwards, sessions of 25 min were recorded where the mice were left to freely behave in the arena. To encourage a better coverage, three chocolate flakes were placed uniformly distributed in the central part of the arena prior to the experiment.

Mouse behavior was recorded at 60 frames per second by a CMOS camera (Basler acA2000-165umNIR) equipped with wide angle lens (CVO GM24514MCN, Stemmer Imaging) that was placed centrally 35 cm below the arena. Three infrared light sources (LIU780A, Thorlabs) were placed 70 cm away from the center, providing homogeneous illumination of the recording arena from below. All recordings were performed at dim room light conditions.

To extract behavioral pose, six virtual markers were placed on relevant bodyparts (nose, tailroot, paws) in 650 uniformly picked video frames from 14 videos. A residual neural network (ResNet-50) was trained to assign the virtual markers to every video frame[13]. The resulting training error was 2.14 pixels and the test error 2.51 pixels, respectively.

To obtain an egocentric time series of $(x, y)$ marker coordinates we aligned the animal video frames from its allocentric arena coordinates to its egocentric coordinates. In order to get a tail to nose orientation from left to right we compute a rotation matrix and rotate the the resulting frame around the center between nose and tail. This results into egocentrically aligned frames and marker coordinates $\mathbf{X} \in \mathbb{R}^{N \times m}$ for each animal, where $N$ represents the recording length, i.e., 90,000 frames and $m = 10$ the $x, y$ marker coordinates. Note that due to the egocentric alignment the x-coordinate for the nose and tail are fixed lines and therefore do not carry any behavioral information. We removed them from the resulting trajectory. Hence the resulting dimensionality of $m$ is equal to 10 (while the original DLC input time series has a dimensionality of 12 per frame). To fit our machine learning model we randomly sampled subsequences $\mathbf{x}_i \in \mathbb{R}^{m \times w}$ from $\mathbf{X}$ that represent 500 ms of behavior i.e., $w = 30$ video frames. In the same manner, we created $\tilde{\mathbf{x}}_{i+1} \in \mathbb{R}^{m \times v}$ that stores the $v = 15$ subsequent time points of $\mathbf{x}_i$ to train the prediction decoder.

**Variational animal motion embedding**. Given a set of $n$ multivariate time series $\mathbb{X} = \{\mathbf{X}^1, \mathbf{X}^2, \ldots, \mathbf{X}^n\}$, where each time series $\mathbf{X}^i = (x^1, x^2 \ldots, x^N)$ contains $N \times m$ ordered real values, the objective of our model is to learn a latent space $\mathbf{Z}$ that captures the dynamics of the time series data. To achieve this goal the multivariate time series $\mathbf{X}^i$ are sampled into defined subsequences $\mathbf{x}_i \in \mathbb{R}^{m \times w}$, where $m$ representing the $x, y$ egocentric marker coordinates and $w$ representing the sampled time window. Now, for every $\mathbf{x}_i$ we learn a vector representation $\mathbf{z}_i \in \mathbb{R}^d$, which effectively reduces its dimension ($d < m \times w$). Hence, $\mathbf{z}_i$ is learned via the non-linear mappings $f_{enc}: \mathbf{x}_i \to \mathbf{z}_i$ and $f_{dec}: \mathbf{z}_i \to \tilde{\mathbf{x}}_i$, where $f_{enc}, f_{dec}$ denotes the encoding and decoding process, respectively and is defined by,

$$\mathbf{z}_i = f_{enc}(\mathbf{x}_i). \tag{1}$$

In order to learn the spatiotemporal latent representation our model encoder is parameterized by a two layer bi-directional RNN (biRNN) with parameters $\phi$. Furthermore, our model uses two biRNN decoder with parameters $\theta$ and $\eta$.

The input data is temporally dependent and biRNNs are a natural choice in order to capture temporal dynamics. They extend the unidirectional RNN by introducing a second hidden layer which runs along the opposite temporal order. Therefore, the model is able to exploit information about the temporal structure from the past and future at the same time. Its hidden representation is determined by recursively processing each input and updating their internal state $\mathbf{h}_t$ at each

timestep for the forward and backward path via,

$$\mathbf{h}_t^f = tanh\left(f_\phi(\mathbf{x}_i^t, \mathbf{h}_{t-1})\right), \quad \mathbf{h}_t^b = tanh\left(f_\phi(\mathbf{x}_i^t, \mathbf{h}_{t+1})\right), \quad \mathbf{h}_c = \mathbf{h}_t^f + \mathbf{h}_t^b \qquad (2)$$

where $\mathbf{h}_t^f$ is the hidden information of the forward pass and $\mathbf{h}_t^b$ is the hidden information of the backward pass, $\mathbf{x}_i^t$ is the current time step of the input sequence $\mathbf{x}_i$, $f_\phi$ is a non-linear transition function, and $\phi$ is the parameter set of $f_\phi$. The transition function $f_\phi$ is usually modeled as long short-term memory (LSTM)[57] or gated recurrent unit (GRU)[29]. Here, we use GRUs as transition function in the encoder and decoder.

The joint probability of a subsequence $\mathbf{x}_i$ is factorized by a RNN as product of conditionals,

$$p_\phi(\mathbf{x}_i) = \prod_{t=1}^{T} p_\phi(x_t | x_{1:t-1}). \qquad (3)$$

In order to learn a joint distribution over all variables, or more precise, the underlying generative process of the data, we apply the framework of variational autoencoders (VAE) introduced by[17,18]. VAEs have been shown to effectively model complex multivariate distributions and can generalize much better across datasets.

**Variational autoencoder.** In brief, by introducing a set of latent random variables $\mathbf{Z}$ the VAE model is able to learn variations in the observed data and can generate $\mathbf{X}$ through conditioning on $\mathbf{Z}$. Hence, the joint probability distribution is defined as,

$$p_\theta(\mathbf{X}, \mathbf{Z}) = p_\theta(\mathbf{X}|\mathbf{Z})p_\theta(\mathbf{Z}), \qquad (4)$$

parameterized by $\theta$.

Determining the data distribution $p(\mathbf{X})$ by marginalization is intractable due to the non-linear mappings between $\mathbf{X}$ and $\mathbf{Z}$ and the integration of $\mathbf{Z}$. In order to overcome the problem of intractable posteriors the VAE framework introduces an approximation of the posterior $q_\phi(\mathbf{Z}|\mathbf{X})$ and optimizes a lower-bound on the marginal likelihood,

$$\log p_\theta(\mathbf{X}) \geq \mathbb{E}_{q_\phi(\mathbf{Z}|\mathbf{X})}[\log p_\theta(\mathbf{X}|\mathbf{Z})] - KL(q_\phi(\mathbf{Z}|\mathbf{X})||p_\theta(\mathbf{Z})), \qquad (5)$$

where $KL(Q||P)$ denotes the Kullback–Leibler divergence between two probability distributions $Q$ and $P$. The prior $p_\theta(\mathbf{Z})$ and the approximate posterior $q_\phi(\mathbf{Z}|\mathbf{X})$ are typically chosen to be in a simple parametric form, such as a Gaussian distribution with diagonal covariance. The generative model $p_\theta(\mathbf{X}|\mathbf{Z})$ and the inference model $q_\phi(\mathbf{Z}|\mathbf{X})$ are trained jointly by optimzing Eq. (5) w.r.t their parameters. Using the *reparameterization trick* (Eq. (6)), introduced by[17] the whole model can be trained through standard backpropagation techniques for stochastic gradient descent.

**Variational lower bound of VAME.** In our case, the inference model (or encoder) $q_\phi(\mathbf{z}_i|\mathbf{x}_i)$ is parameterized by a biRNN. By concatenating the last hidden states of the forward and backward steps of the biRNN we obtain a global hidden state $\mathbf{h}_i$, which is a fixed-length vector representation of the entire sequence $\mathbf{x}_i$. To get the probabilistic latent representation $\mathbf{z}_i$ we define a prior distribution over the latent variables $p_\theta(\mathbf{z}_i)$ as an isotropic multivariate Normal distribution $\mathcal{N}(\mathbf{z}_i; \mathbf{0}, \mathbf{I})$. Its parameter $\mu_z$ and $\Sigma_z$ of the approximate posterior distribution $q_\phi(\mathbf{z}_i|\mathbf{x}_i)$ are generated from the final encoder hidden state by using two fully connected linear layers. The latent representation $\mathbf{z}_i$ is then sampled from the approximate posterior and computed via the reparameterization trick,

$$\mathbf{z}_i = \mu_z + \sigma_z \odot \epsilon, \qquad (6)$$

where $\epsilon$ is an auxiliary noise variable and $\odot$ denotes the Hadamard product.

The generative model $p_\theta(\mathbf{x}_i|\mathbf{z}_i)$ (or decoder) receives $\mathbf{z}_i$ as input at each timestep $t$ and aims to reconstruct $\mathbf{x}_i$. We use the mean squared error (MSE) as reconstruction loss, defined by,

$$\mathcal{L}_{MSE} = \frac{1}{n} \sum_{i=1}^{n} ||\mathbf{x}_i - \tilde{\mathbf{x}}_i||_2^2. \qquad (7)$$

The log-likelihood of $\mathbf{x}_i$ can be expressed as in Eq. (5). Since the KL divergence is non-

---

negative the log-likelihood can be written as

$$\mathcal{L}(\theta, \phi; \mathbf{x}_i) = \mathbb{E}_{q_\phi(\mathbf{z}_i|\mathbf{x}_i)}[\log p_\theta(\mathbf{x}_i|\mathbf{z}_i)] - KL(q_\phi(\mathbf{z}_i|\mathbf{x}_i)||p_\theta(\mathbf{z}_i)). \qquad (8)$$

Here, $\mathcal{L}(\theta, \phi; \mathbf{x}_i)$ is a lower bound on the log-likelihood and therefore called the *evidence lower bound* (ELBO) as formulated by[17].

We extend the ELBO in our model by an additional prediction decoder biRNN $p_\eta(\tilde{\mathbf{x}}_i|\mathbf{z}_i)$ to predict the evolution $\tilde{\mathbf{x}}_i$ of $\mathbf{x}_i$, parameterized by $\eta$. The motivation for this additional model is based on[30], where the authors propose a composite RNN model which aims to jointly learn important features for reconstruction and predicting subsequent video frames. Here, $p_\eta(\tilde{\mathbf{x}}_i|\mathbf{z}_i)$ serves as a regularization for learning $\mathbf{z}_i$ so that the latent representation not only memorizes an input time series but also estimates its future dynamics. Thus, we extend Eq. (8) by an additonal term and parameter,

$$\mathcal{L}(\theta, \phi, \eta; \mathbf{x}_i) = \mathbb{E}_{q_\phi(\mathbf{z}_i|\mathbf{x}_i)}[\log p_\theta(\mathbf{x}_i|\mathbf{z}_i)] + \mathbb{E}_{q_\phi(\mathbf{z}_i|\mathbf{x}_i)}[\log p_\eta(\tilde{\mathbf{x}}_i|\mathbf{z}_i)] \\ - KL(q_\phi(\mathbf{z}_i|\mathbf{x}_i)||p_\theta(\mathbf{z}_i)). \qquad (9)$$

Finally, the training objective is to minimize

$$\min_{\theta, \phi, \eta} \mathcal{L}(\theta, \phi, \eta; \mathbf{x}_i) \qquad (10)$$

and the overall loss function can be written as

$$\mathcal{L}_{total} = \mathcal{L}_{reconstruction} + \mathcal{L}_{prediction} + \mathcal{L}_{KL}, \qquad (11)$$

where $\mathcal{L}_{prediction}$ is the MSE loss of the prediction decoder.

The full model was trained on the combined dataset ($1.3e6$ time points) using the Adam optimizer[58] with a fixed learning rate of 0.0005 on a single Nvidia 1080ti GPU. All computing was done with PyTorch[59]. The ergodic mean of the reconstruction error $\mathcal{L}_{reconstruction}$ for all virtual marker time series was found to be 1.82 pixels.

**Motif identification.** To determine the set of behavioral motifs $B = \{b_1, ..., b_K\}$ we obtained the latent vector $\mathbf{Z}$ from a given dataset using VAME as described in Methods 4.10. Here, $\mathbf{Z} \in \mathbb{R}^{d \times N - w}$, with the embedding dimension $d$, the number of frames $N$ and the temporal window $w$, represents the feature space from which we want to identify the motif structure. By treating the underlying dynamical system as a discrete-state continuous-time Markov chain, we apply a Hidden-Markov-Model (HMM) with a Gaussian distributed emission probability to this space to detect states (motifs). We used the **hmmlearn** python package in our framework to implement the HMM. The default settings for the Gaussian emission model from the packages were used. Moreover, we compared the HMM to a simpler and less time consuming k-means clustering in Supplementary Section 1. To identify the number of motif present in our dataset, we used a similar approach as in[7]. We let a HMM infer 100 motifs and identified the threshold, where motif usage dropped below 1%, see Supplementary Fig. 8. Motif usage was determined as the percentage of video frames that are assigned to the occurrence of a specific motif.

To model the transitions between behavioral motifs, we interpreted the motif sequence as a discrete-time Markov chain where the transition probability into a future motif is only dependent on the present motif. This results in a $K \times K$ transition probability matrix $\mathcal{T}$, with the elements

$$\mathcal{T}_{lk} = P(b_k|b_l), \qquad (12)$$

being the transition probabilities from one motif $b_l \in B$ to another motif $b_k \in B$, that are empirically estimated from clustering of $\mathbf{Z}$.

In order to obtain a hierarchical representation of behavioral motifs we can represent the Markov chain (12) as a directed graph $\mathbb{G}$ consisting of nodes $v_1 ... v_K$ connected by edges with an assigned transition probability $\mathcal{T}_{lk}$. We can transform $\mathbb{G}$ into a binary tree $\mathbb{T}$ by iteratively merging two nodes $(v_i, v_j)$ until only the root node $v_R$ is left. Every leaf of this tree represents a behavioral motif. To select $i$ and $j$ in each reduction step, we compute the cost function

$$C_R = \min_{i,j}\left(\sum_{i,j} \frac{U_i + U_j}{\mathcal{T}_{ij} + \mathcal{T}_{ji}}\right), \qquad (13)$$

where $U_i$ is the probability of occurrence for the $i$th motif. After each reduction step the matrix $\mathcal{T}$ is recomputed in order to account for the merging of nodes. Lastly, we obtain *communities* of behavioral motifs by cutting $\mathcal{T}$ at given depth of the tree, analogous to the hierarchical clustering approach used for dendrograms. Note that the cost function is chosen to allow for the detection of the most highly connected motifs first but in different settings this might be altered to achieve good results.

**Manually assigned labels and scoring.** In order to obtain manually assigned labels of behavioral motifs we asked three experts to annotate one recording of freely moving behavior with a duration of 6 min. All three experts had a strong experience in vivo experiments as well as ethogram-based behavior quantification. The experts could scroll trough the video in slow-motion forward and backward in time and annotated the behavior into several atomic motifs as well as a composition of those. As an example, the experts were allowed to annotate a behavioral sequence as *walk* or *exploration*, but also *walk and exploration*. We then summarized the annotation into atomic motifs into 5 coarse behavioral labels, as shown in Table 2.

---

**Table 2 Assignment of atomic motifs into coarse behavior labels.**

| Coarse label | Assigned atomic motif |
| --- | --- |
| Walk | Walk, walk and bend, walk and sniff |
| Pause | No locomotion, Bending, looking up or down while standing still |
| Groom | Groom |
| Rear | Rear, low-rear, wall-rear |
| Exploratory | Undirected sniffing while standing still, bending, looking up or down |

**Table 3 Mouse ethology database taxonomy corresponding for each manually assigned coarse label.**

| Coarse label | Mouse Ethogram database |
|---|---|
| Walk | Active behavior - General activity - Exploratory behavior - Search - General locomotion |
| Pause | Inactive behavior- Still and alert |
| Groom | Active behavior - Maintenance behaviors - Grooming |
| Rear | Active behavior - General activity - Exploratory behavior - Search - Rearing |
| Exploratory | Active behavior - General activity - Exploratory behavior - Investigate - Undirected sniffing |

The coarse labels were created with respect to the behavior descriptions taken from the Mouse Ethogram database (www.mousebehavior.org), which provides a consensus of several previously published ethograms. The assignment of coarse labels to the Mouse Ethogram database taxonomy is shown in Table 3.

For scoring of human assigned labels to VAME motifs we used the clustering evaluation measures Purity, NMI and Homogeneity. Purity is a measure of the extent to which clusters contain a single class. NMI (from *scikit-learn*) is a normalization of the Mutual Information score to scale the results between 0 (no mutual information) and 1 (perfect correlation). Homogeneity is, in its essence, a more strict Purity measure. From *scikit-learn*: A clustering result satisfies homogeneity if all of its clusters contain only data points which are members of a single class. Purity is defined as

$$\text{Purity}(U, V) = \frac{1}{N} \sum_{u \in U} \max_{v \in V} |u \cap v|, \quad (14)$$

where $U$ it the set of manually assigned labels $u$, $V$ is the set of labels generated by VAME $v$ and $N$ is the number of frames in the behavioral video. The Normalized Mutual Information score is written as

$$\text{NMI}(U, V) = \frac{\text{MI}(U, V)}{E(H(U), H(V))}, \quad (15)$$

where $\text{MI}(U, V)$ is the mutual information between set $U$ and $V$ defined as

$$\text{MI}(U, V) = \sum_{u \in U} \sum_{v \in V} \frac{|u \cap v|}{N} \log\left(\frac{N|u \cap v|}{|u||v|}\right), \quad (16)$$

and $H(U)$ is the entropy of set $U$ defined as

$$H(U) = -\sum_{i=1}^{|U|} \frac{|u \cap v|}{N} \log\left(\frac{|u \cap v|}{N}\right), \quad (17)$$

where the $||$ operator denotes the amount of frames that have the corresponding labels assigned. Homogeneity is defined as

$$\text{Homogeneity} = 1 - \frac{H(U|V)}{H(U)}, \quad (18)$$

where the conditional entropy of manually assigned labels given the cluster assignments from VAME is given by

$$H(U|V) = -\sum_{u=1}^{|U|} \sum_{k=1}^{|K|} \frac{u \cap v}{|u \cap v|} \log\left(\frac{u \cap v}{|v|}\right), \quad (19)$$

Note that the Purity score (14) is larger when the set $V$ is larger than $U$ and the NMI score (15) is generally larger when both sets $U$ and $V$ are of similar size, i.e., the number of possible labels is roughly the same in the human assigned set as well as the set generated using VAME.

**Human phenotype classification task**. For the classification of phenotypes using human experts we have created an online form, where experts could watch all eight videos and make their choice about which phenotype is shown in each video. There was no time limit and the average time to complete the questionnaire was 30 min. The participants have not been told how many animals of each group are in the set. For every video, the following five decision could be made: APP/PS1 (Very sure), APP/PS1 (Likely), Unsure, Wildtype (Likely), Wildtype (Very Sure). We have counted a right answers (*Very sure* and *Likely*) as a correct classification (1 point), and wrong answers as well as the choice for the *Unsure* option as wrong classification (0 points). Eleven experts were participating in this classification task. All of them had previous experience with behavioral video recordings in an open field and/or treadmill setting. In addition, six of the participants had previous experience with the APP/PS1 phenotype.

**Statistics and reproducibility**. Statistical analysis was performed using Prism 8. To quantify motif usage between transgenic ($n = 4$) and wildtype ($n = 4$) mice a multiple t-test was used and statistical significance was determined using the Holm–Sidak method. Multiple comparison 2-way ANOVA was used for the transition statistics with post-hoc Sidak's multiple comparison test. To evaluate statistical significance in the locomotor activity of transgenic and wildtype mice data were subjected to an unpaired *t*-test. All data presented are shown as mean ± standard deviation and the threshold for significance was set at $p < 0.05$.

**Reporting summary**. Further information on research design is available in the Nature Research Reporting Summary linked to this article.

## Data availability

The data used within this manuscript is available at https://figshare.com/articles/media/VAME_Data/19213272.

## Code availability

The VAME toolbox is open-source and available to the scientific community at https://github.com/LINCellularNeuroscience/VAME[60].

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

## Acknowledgements
We thank J. Macke, E. Restrepo, J. Gall and S. Stober for comments on the manuscript. This work was supported by the European Research Council (CoG;SUBDECODE), NIA P01AG073082 (J.J.P.) and DFG-SFB 1436 and 1089.

## Author contributions
Conceptualization, K.L., P.B., and S.R.; methodology, K.L. P.M., S.M., and P.B.; experiments, K.L.; code development, K.L., J.K., and P.B.; experimental design, K.L., P.B., and F.F.; writing/editing, K.L., P.M., P.B., and S.R.; reviewing, K.L., P.M., P.B., S.M., J.P., and S.R.; supervision, P.B. and S.R.; funding acquisition. S.R.

## Funding

## Competing interests
The authors declare no competing interests.
