## [Peer Review File · Communications Biology]

Reviewers' comments:

Reviewer #1 (Remarks to the Author):

This paper proposes a novel framework for clustering and labelling motifs of mouse behaviour using a temporal VAE, trained on the output of DeepLabcut, followed by a Hidden Markov Model to cluster the motifs.

My expertise is machine learning, not animal modelling so, I can only confidently review this aspect. However, the paper is well motivated and the results look encouraging - clearly the between population comparisons of motif expression is able to pick up behavioural differences between the phenotypes.

My problem is with the presentation of the paper, which contributes to the model coming across as quite overcomplicated. In general, I found the explanation of the methods split between the results section 2.1 and the methods section 4 extremely confusing and have had to go back and forth constantly to cross check the dimensionality of the variables. Some of the notation was quite confusing.

For example in sec 2.1 it states the input to the model is the x-y coordinates across time for 6 landmarks. As such, X should have shape 90000x12; however in section 2.1 it states $m=10$ not 12. Is this related to the creation of the egocentric coordinate system?

After that the data is passed through a VAE to generate a latent space with dimension d , where the full dimension of z (defined in section 4.4) is $d \times (N-T)$. This makes it appear that the VAE is expanding the spatial dimensions from m to d (10 to 30) but in 2.1 it instead states that each z_i is a compression of time windowed samples x_i of shape $m \times \omega$. However, ω is not mentioned at all in section 4. Also what is n (the set of multivariate time series) - is this the number of samples you get after time windowing?

From what I can tell the use of N and n are perhaps getting mixed up here? Or perhaps n and N , since if $X = \{x_1 x_2 x_3 \dots x_n\}$ then surely $Z = \{z_1 z_2 z_3 \dots z_n\}$ and Z should have shape $d \times n$? Maybe $n = N - T$? Either way the notation and the description of the methods really need clarifying and making consistent. The workflow of figure 1 could be made clearer by perhaps clarifying the dimensionality of inputs and outputs there.

As a result of this confusion it became quite unclear why the VAE was needed at all. Would it not be possible to just pass X directly to the HMM? If so the impact of the VAE should be tested in an ablation experiment and the results should be added to table 2.

Finally there are number of other minor notational or typo issues that I list below:

Pg 11 equation (2) needs spacing out;

Pg 11 below (2) f should I think be f_{ϕ} . Its not the most helpful notation since you also have f for 'forward' (as for $h_f t$)

Pg 11 below (3) typo 'distirbutions'

Pg 12 'concatenating the hidden states of the forward and backward path we obtain' - are these the forward and backward steps of the biRNN? Its not clear

Pg 12 typo log-likelihood

Reviewer #3 (Remarks to the Author):

Luxem et al provide a study comprised in two parts. The first is the description of a behavioural segmentation algorithm using VAE. The second is a comparison of a transgenic Alzheimer's transgenic mouse line to wildtypes. There is plenty of quality here, and the scientists involved demonstrate their meticulousness.

My major concern however is about if/how the two parts come together. The whole of the introduction is about recent computational methods and ethological methods. Section 2.1 and Fig 1 follow suit. The paper then switches somewhat abruptly to quantifying the tg differences for fig 2 and 3. Moreover, while fig 3 uses the output of VAME, all of this work could be done with another segmenting method (see comments). I'll also say that fig 3 was the most convincing interesting and well done part of the paper for me. Fig 4 then returns abruptly to VAME quantification, and it is assumed that these data are from WT animals.

As a document showcasing the innovation and benefits of VAME, the paper falls short. But I think this is easily salvageable. The first manner would be to expand, going into sufficient detail about the algorithmic benefits of the approach (why did you make X decision, why use this or that) and operationalizing the method for potential users. Additionally, it would benefit the authors to show another example or go deeper. For instance, VAME produces a handful of walking, rearing, etc types that show differences in tg, but I really don't know why these are important/beneficial to the comparison or how they'd help my work in a related project.

The other option is much easier I'd think. Repackage the paper as a study of AD that provides much better resolution with this new technique. Some retooling of the intro, layout, etc may be necessary (and title?), but it wouldn't require as much effort. Here again though, telling us WHY something is interesting/important is vital (the AD tg mouse does more of X, but how would a translational rodent scientist or clinician benefit from knowing this information?!) Given the success of the biorxiv version, this may make more sense?

Either way, as a reviewer I find that this reads more akin to an arxiv summary than a manuscript - there's just not sufficient depth and explanation. I trust in the authors to be able to solve these issues, laid out more clearly below, but the current manuscript does not have the required depth.

Overall / methods - there's a lot of missing information on the code. What are the requirements? Can it work on SLEAP? We have some basics on how the model was created for this paper (1.3e6 timepoints, a 1080ti gpu), but for the potential user, there's no information on how to operationalize this method. If this is a new method to solve the AD characterization of this paper - great. If instead this is a tool to be added to a behaviorist's toolset, it's just not there.

p2 - Could the authors please clarify " the extraction of underlying discrete states as a basis for quantification". it may really help them to clarify what DLC, SLEAP can't do, as well as things like simba or b-side.

p2 - Similarly, I know what VAME does and am very familiar with the code. Most users won't be. More importantly, most won't understand what benefits or use cases VAME might outperform other options and also have limited computational background. Even with a math background, reading the paragraph on what VAME's setup allows leaves it unclear why I'd use this.

pg 2 - The animal was aligned to the nose-tail center. occasionally the nose will be very off to the side or even behind the animal. Please state in the text if this should have any effect

pg 2 - explaining the basics of the math is very clear to someone with a bit of background. thank you for this

pg 4 - Please unpackage " specific (repetitive) behavioral task settings"

pg 4 - The idea of trail is introduced here, but never defined

p5 - it's unclear to me why the thorough quantification of human rater is valuable

fig 2 / general. There are 4 animals in each of two groups. While the overall clustering make sense, we don't get a good idea of what the difference between 44 and 42 are.. . yet one is highly sig, the

other is not. The reader would be at much more of an advantage if some sort of kinematic readout was provided to get a sense of things. It may be that the tg animals just move more. This is fairly evident in Table 1, and a strong trend in 2A, and one that I'm guessing would be resolved with more animals. As such, it is difficult to interpret how this relationship lays onto the actions. If I were to randomly subsample unsupported rearing in these two modest samples, what is the probability of seeing a distinction? This gets at the heart of "what is VAME attaching to". If it's just a simple kinematic parameter, these results are valid but not particularly interesting. If VAME can see something deeper, there is real use to it

I also want to say, if the goal is identifying "behavioral differences that would be undetected by human observation" - as Ann Kennedy and dozens of scientists before her have pointed out, human annotators are really quite poor, so this is a low bar. Could a simple ethoVision readout / just the distance traveled by the head give the same level of discernment? I'd much rather a slightly more intelligent strawman.

- I may have missed it, but once trained, can this VAME model be applied to other tg/wt datasets? It seems, particularly with the small-ish n and lack of details to why a certain number of groups were selected, that this could be a problem. e.g. by random happenstance, I can get 3 unsupported walking groups and you get 4, and the within-action difference you see may not exist in my 3.

fig 3 - I LOVED figure 3 and think we need more of it. This sort of analysis should be the future of computational neuroethology and movement study.

p7 - "On the community level, the temporal structure of behavior can be identified by observing the probability of a community transitioning into another" - It is unclear to me how this differs from saying "we can compare a walk to other walks, or to all behaviours". Apologies if I've lost the thread, but to me this is not inherent to VAME's scales or temporal structure. Most any method, even supervised, can subdivide larger communities into discrete sub-categories. If the rationale is because VAME does so in a mathematically rigorous fashion, then we now have problems with figure 4. It assumes essentially identical groupings from a qualitative/historical background. Said more simply, please help me understand why this would be a feature specific to VAME

Fig 4 - I applaud the authors for attempting this comparison across methods. It's not a fun or comfortable thing to do, but hopefully is helpful to the field in the end. It's also difficult to compare methods given a lack of ground truth. The authors go to some effort earlier in the manuscript to point out how poorly human raters are, and that while there is no ground truth to behaviour, humans certainly shouldn't be trusted to come close to it. I would not consider 72% to be a good score for 3/3 agreement.

Some questions:

- many details on the non-VAME methods are missing. Same DLC data used (I assume)? How many motifs were there? It seems as if there are 50 for all, which I'd assume means the other methods were forced to do so. If that is the case, this is unfair. One technique may be better at communities while another motifs - and thus we often see valid approaches with 10 vs 50 vs 100 motifs. Even if all methods aligned to 50, I'd be curious as a potential user about how the performance varied by category. This can be taken in at least 2 ways. If one technique hones in on motion energy, it may oversplit/undersplit compared to other methods that are internally consistent but don't match the human. More to the point, if I'm doing a study of rearing, I may not care about the performance in the other categories. You have plenty of text to spare. Tell us more!

- I'm not following something, and my apologies if that applies more broadly. This is a complicated plot, so some more explanation might be helpful. Humans rated each frame with one of 5 behavioral labels. For 4c, blue indicates how much inter-rater agreement there was on each frame that was labeled motif 0-49, yes? Either way, just a bit of text will help understanding and ultimately adoption of the technique. There is plenty of opportunity for these approaches to be unfairly applied. Provide detail and make it clear to us that this is not the case.

Somewhat related, an n of 4 x 2 groups is fine, but with counts more similar to those typically used in AD mouse studies, I'd expect at least 5x that. Additionally, I'd appreciate some discussion of the

empty columns. VAME has 5, while the others have 2 and 3. Could you explain for the reader what having 5 missing motifs might do to the scores in the table? Ostensibly you're comparing performance across fewer groups, which should boost VAME above the others, though it likely depends on how this is done.

p9 - " MotionMapper may be more suitable for the detection of motifs in fruit flies." I fully agree, could you please state why? I assume this comes from the reliance on motion energy. Making a clear statement about moseq would also be good here.

p9 - " All these approaches require deep, reliable and complete dissection of behavioral motifs." Absolutely! And I would LOVE to see the quantification of this in the document. This is somewhat similar to the question above about generalization, but I honestly have no idea how reliable (or complete?) VAME is. Canonical categories have multiple motifs (which is common to most if not all techniques), and VAME performs more akin to humans... though these motifs are ignored. This ignoring is reasonable, but I really don't know what the motifs mean.

p10 - " This approach, however, does not yield temporal information about the behavior" I'm not sure that I agree, at least when stated in such broad terms. I believe it uses speed (distance/time) and certainly provides the time of each action. Could the authors please clarify?

p10 / general. From the first sentence to the discussion, behavioural dynamics is mentioned, but it's not clear what the author mean. Kinematics are notably absent, and transitions can be done with any approach. I think there is something real here for the authors to capitalize on, but it's not clear to me.

Minor

- page 1 - I'd assert that subsecond scale isn't the critical aspect here, and also that such an assertion is wrong (it may find larger motifs, and the eye can detect quick behaviors like tics with ease)
- page 2 - To avoid confusion when discussing pose estimation, you may want to use "computer vision" in place of "supervised deep learning"
- 1e, the text should be a bit larger. Also, if the text states 1A or 1B, the figure should not have a and b. Lastly, it looks like many things are listed as 1A but should be b,c,d...
- page 7 - while " Purity, Normalized Mutual Information (NMI) and Homogeneity" are all mathematically defined deep within the methods, I'm sure it would help many reader to get a hand-wavy explanation of these terms. Similarly, why NMI instead of AMI? Not too important, but the newer AMI metric has the benefit of being normalized to chance.
- page 7 - I'd change " In order to validate all three models (VAME, AR-HMM, MotionMapper) we created..." to "To validate three models (VAME, AR-HMM used in MoSeq, MotionMapper) we created..."
- I appreciate the correct spelling of behaviour in some parts of the manuscript, but it is inconsistent.

Reviewer #4 (Remarks to the Author):

Summary:

This excellent submission details a new framework (VAME) for identifying hierarchical structure from recordings of animal behavior. The authors demonstrate the utility of their approach by identifying (biologically relevant) differences between transgenic and wild-type animals, and show that VAME performs favorably on a benchmark dataset when compared to similar approaches.

I am convinced that the paper will be of substantial interest to the community; particularly given the exceptionally well documented code provided on github. Indeed, the communities' interest is already clear from the > 30 citations the associated preprint has received and almost 100 stars on github. Relatedly, by providing high quality code and detailed parameter choices (Table S.3) the authors maximize the potential reproducibility of their work.

I would, however, like to suggest a number of minor changes which would strengthen the manuscript.

Minor Changes:

1. In section 2.1 it is not clear how "six virtual markers" (4 paws, nose, tailbase) yield "10 (x, y) - marker positions".
2. In section 2.1 and the first paragraph of the discussion the authors the phrase "richer representation", which is vague. The authors should describe the difference in the representation with and without the additional biRNN decoder, and ideally add results to the supplement detailing this comparison.
3. While some choices regarding hyperparameters are always necessary, it would be nice if the authors could better justify their choice of 30 dimensions for their latent representation (section 2.2).
4. Related to point 3, the authors should discuss if the same latent dimension size is appropriate for different animals or for animals with different genetic backgrounds (i.e., wild-type / transgenic).
5. In section 2.2, the authors should make it clearer that they identify the same 50 motifs in each animal (as opposed to 50 unique motifs per animal or a mixture of shared and unique motifs).
6. In section 2.2 the authors should highlight that motif usage is highly similar (has low variance) across animals. This would support their claim that VAME identifies robust / common behaviors.
7. However, following point 6, it is not clear to me if the author's approach would be able to identify motifs used by only one population (here half of the animals). Such motifs would be of interest as they would represent significant changes in behavior.
8. Figure S.1 (particularly panels A, B and C) provide an intuitive graphical description of the author's comparison between wild-type and transgenic animals. The manuscript would be improved if the authors could either incorporate some aspects of S.1 into Figure 2, or swap table 1 and S.1. I acknowledge that this may be preference of style.
9. In Figure 2B the authors convincingly demonstrate that human observers cannot classify wild-type vs transgenic animals with greater than chance accuracy. The authors should extend this result by showing that VAME (presumably) enables greater classification accuracy based on motif usage. One approach to this would be to use linear discriminant analysis classifiers. In Ghosh and Rihel 2020 (eNeuro), for example, the authors demonstrate that differences in motif usage can be used to accurately distinguish larval zebrafish behavior across a variety of contexts, such as exposure to different concentrations of a compound (Figure 5A and B).
10. Related to 9, a comparison of the motif usage-based classification accuracy (wild-type vs transgenic) across VAME, AR-HMM and MotionMapper would be an interesting addition to the paper, as it would determine which method is best able to detect a biologically relevant difference.
11. The comparison between manual annotators and methods is convincing (Table 2), however a graph of these values as a function of k would strengthen the authors claim.
12. In the discussion the examples of supervised approaches are repeated: "Supervised approaches like DeepEthogram, SimBa or MARS DeepEthogram, SimBa or MARS".
13. In general it would be good if the authors could better motivate their use of a bottom-up camera, and discuss what one may hope to discover from applying VAME to existing datasets acquired from this perspective. For example, Darmohray ... and Carey 2019 (Neuron).

14. The general appeal of the paper would be increased if the authors could expand upon how VAME could be applied to other types of data.

15. Finally, the text accompanying supplement 5.8 (Generative model aspects) would be improved with more interpretation of the figure. Currently, it is not clear what the take away is, or more precisely how similar the reconstructions are to the generated samples.

Point-by-Point response letter

We thank all three reviewers for their time and work invested into reviewing our manuscript. Below we address each question and concern raised by the respected reviewer and highlight changes within our manuscript.

Reviewer #1 (Remarks to the Author)

This paper proposes a novel framework for clustering and labelling motifs of mouse behaviour using a temporal VAE, trained on the output of DeepLabcut, followed by a Hidden Markov Model to cluster the motifs.

My expertise is machine learning, not animal modelling so, I can only confidently review this aspect. However, the paper is well motivated and the results look encouraging - clearly the between population comparisons of motif expression is able to pick up behavioural differences between the phenotypes.

My problem is with the presentation of the paper, which contributes to the model coming across as quite overcomplicated. In general, I found the explanation of the methods split between the results section 2.1 and the methods section 4 extremely confusing and have had to go back and forth constantly to cross check the dimensionality of the variables. Some of the notation was quite confusing.

We agree with the reviewer that our notation needed to be cleaned up. Hence, we revised the manuscript to have the same notation throughout the sections and made the dimensions explicit.

For example in sec 2.1 it states the input to the model is the x-y coordinates across time for 6 landmarks. As such, X should have shape 90000×12 ; however in section 2.1 it states $m=10$ not 12. Is this related to the creation of the egocentric coordinate system?

We addressed this by updating the text in our method section 4.3 and having now a reference to this section in section 2.1. This is indeed a result of the egocentric alignment as two of the marker points are used as anchors for alignment, after the other points are rotated around them they are set to zero. Thus, we remove the two dimensions after the egocentric alignment.

“Note that due to the egocentric alignment the x-coordinate for the nose and tail are fixed lines and therefore do not carry any behavioral information. We removed them from the resulting trajectory. Hence the resulting dimensionality of m is equal to 10 (while the original DLC input time series has a dimensionality of 12 per frame).” (line 603 - 606)

After that the data is passed through a VAE to generate a latent space with dimension d , where the full dimension of z (defined in section 4.4) is $d \times (N-T)$. This makes it appear that the VAE is expanding the spatial dimensions from m to d (10 to 30) but in 2.1 it instead

states that each z_i is a compression of time windowed samples x_i of shape $m \times \omega$. However, ω is not mentioned at all in section 4. Also what is n (the set of multivariate time series) - is this the number of samples you get after time windowing?

We agree with the reviewer that we did not make this clear enough and updated our notations to make the dimension explicit. Moreover, we updated section 4.4 (line 610 - 614) to describe the process of dimensionality reduction in more detail and we made sure to use the same notation, hence w is now also referred to in section 4.5. For clarification, VAME takes in a trajectory sample of dimension $(m \times w)$ and projects this into a latent vector z of dimension $d < (m \times w)$, effectively reducing its dimensionality. We also now clarified this in section 2.1 (line 90 - 91). Moreover, the small letter n is used for the number of multivariate time series (line 610). This means we have n multivariate time series with each about $N=90000$ frames in our experimental dataset.

From what I can tell the use of N and are perhaps getting mixed up here? Or perhaps n and N , since if $X=\{x_1 x_2 x_3 \dots x_n\}$ then surely $Z=\{z_1 z_2 z_3 \dots z_n\}$ and Z should have shape $d \times n$? Maybe $n=N-T$? Either way the notation and the description of the methods really need clarifying and making consistent. The workflow of figure 1 could be made clearer by perhaps clarifying the dimensionality of inputs and outputs there.

We cleaned our notation throughout the manuscript to make it consistent and cleared up these notational ambiguities. We also updated the dimensions of Figure 1 as well as some font sizes to improve readability.

As a result of this confusion it became quite unclear why the VAE was needed at all. Would it not be possible to just pass X directly to the HMM? If so the impact of the VAME should be tested in an ablation experiment and the results should be added to table 2.

We agree with the reviewer and added an additional ablation experiment to table 2 and supplemental table S.1. The result demonstrates that passing the egocentric pose data directly into an HMM is significantly lower in performance than our approach, but gives similar performance as the MotionMapper framework. We also added additional text to section 2.4 describing these results (line 230 - 236).

Finally there are number of other minor notational or typo issues that I list below:

Pg 11 equation (2) needs spacing out;

We fixed equation 2 (line 623).

Pg 11 below (2) f should I think be φ . Its not the most helpful notation since you also have f for 'forward' (as for $h_f t$)

We fixed the notation (line 624-626).

Pg 11 below (3) typo 'distirbutions'

We fixed the typo (line 629).

Pg 12 'concatenating the hidden states of the forward and backward path we obtain' - are these the forward and backward steps of the biRNN? Its not clear

We changed the sentence to "By concatenating the last hidden states of the forward and

backward steps of the biRNN we obtain” to make it more readable (line 647- 648).

Pg 12 typo log-likelihood
We fixed the typo (line 657).

Reviewer #3 (Remarks to the Author):

Overall / methods - there's a lot of missing information on the code. What are the requirements? Can it work on SLEAP? We have some basics on how the model was created for this paper (1.3e6 timepoints, a 1080ti gpu), but for the potential user, there's no information on how to operationalize this method. If this is a new method to solve the AD characterization of this paper - great. If instead this is a tool to be added to a behaviorist's toolset, it's just not there.

We understand the point of the reviewer and decided to extend our paper to highlight the use of VAME as a tool that researchers can use to extract states and dynamics from their time series signals (behavior or otherwise). Therefore, we added an additional method section 4.1 “VAME protocol” (from line 373) that addresses the shortcomings mentioned here and also integrates other comments (which we will highlight).

The new method section essentially provides intuition on why to use VAME and walks the reader/user through the installation and workflow process of VAME. We also provide some common pitfalls so that these can be avoided and refer to downstream analysis that users can apply with the data coming from VAME. With this section, we also present an additional Figure 5 and three other supplemental Figures to help to guide users.

We motivate the new method section in section 2.1 (line 111-115):

“We want to make the reader aware that the method section 4.1 provides a full protocol of the VAME workflow with additional information on how to use the method and use it to answer their experimental questions. Here, we discuss the installation of the framework, the necessary pre-processing steps, the training of the model, and the final steps comprising the visualization of the motif time series and the embedding space. Finally, we discuss common pitfalls and direct the reader to further downstream analysis.”

p2 - Could the authors please clarify " the extraction of underlying discrete states as a basis for quantification". it may really help them to clarify what DLC, SLEAP can't do, as well as things like simba or b-side.

We agree with the reviewer that this statement needs clarification. Hence, we re-phrased it to (line 44-45):

”However, while such tools provide a continuous virtual marker signal of the animal body motion, the extraction of underlying dynamics and motifs remains a key challenge.”

Moreover, we discuss the difference to tools like B-SOiD and SimBa on page 11 (line 349-354)

pg 2 - Similarly, I know what VAME does and am very familiar with the code. Most users won't be. More importantly, most won't understand what benefits or use cases VAME might outperform other options and also have limited computational background. Even with a math background, reading the paragraph on what VAME's setup allows leaves it unclear why I'd use this.

Thank you for pointing this out. This is now addressed within the new section 4.1. Here, users are guided with demo data through VAME. Hence, they can understand the output of VAME better and connect this with the first part of the manuscript. Moreover, we give some advice on downstream analysis that can be tackled with VAME.

pg 2 - The animal was aligned to the nose-tail center. occasionally the nose will be very off to the side or even behind the animal. Please state in the text if this should have any effect

Within our new section 4.1.3 we added the following text to clarify this (line 447-451):

“Using two appropriate anchor points for the egocentric alignment is crucial to transform the animal from its allocentric virtual marker coordinates to its own egocentric coordinate system. However, these points can be sometimes occluded by e.g. the animals body. If this happens, the pose estimation will output a low accuracy of the respective key point and the `vame.egocentric_alignment()` function will set the value of this keypoint to NaN ("Not a Number"), which will be later interpolated when creating the training dataset.”

Moreover, related to this, we also added a section 4.1.7 “Pitfalls and downstream analysis” (line 534), where we advise users to carefully check their pose estimation model. Identity switching of virtual markers can have a strong effect within the learned embedding space of VAME and reduces the models capability to identify robust dynamics and motifs.

pg 2 - explaining the basics of the math is very clear to someone with a bit of background. thank you for this

Thank you for this comment.

pg 4 - Please unpackage " specific (repetitive) behavioral task settings”

We agree with the reviewer that this phrase was too unspecific and unpackaged its meaning (line 122 - 125):

“In the past, batteries of behavioral tests were used to assess possible differences between genotypes since no differences could be detected in the open field tests (Giovannetti et al., 2018). Hence, this dataset forms an ideal use-case for the purpose of unsupervised behavior quantification to evaluate whether our proposed method can detect those differences.”

pg 4 - The idea of trial is introduced here, but never defined

Thank you for making us aware, there is of course no trial structure and we performed one single experiment for all animals in which we collected the data. We corrected this by taking out “trial” and replacing it with “experiment” (line 128).

p5 - it's unclear to me why the thorough quantification of human rater is valuable
fig 2 / general. There are 4 animals in each of two groups. While the overall clustering make sense, we don't get a good idea of what the difference between 44 and 42 are.. . yet one is highly sig, the other is not. The reader would be at much more of an advantage if some sort of kinematic readout was provided to get a sense of things. it may be that the tg animals just moves more. This is fairly evident in Table 1, and a strong trend in 2A, and one that I'm guessing would be resolved with more animals. As such, it is difficult to interpret how this relationship lays onto the actions. If I were to randomly subsample unsupported rearing in these two modest samples, what is the probability of seeing a distinction? This gets at the heart of "what is VAME attaching to". If it's just a simple kinematic parameter, these results are valid but not particularly interesting. If VAME can see something deeper, there is real use to it I also want to say, if the goal is identifying "behavioral differences that would be undetected by human observation" - as Ann Kennedy and dozens of scientists before her have pointed out, human annotators are really quite poor, so this is a low bar. Could a simple ethoVision readout / just the distance traveled by the head give the same level of discernment? I'd much rather a slightly more intelligent strawman.

We understand the concern of the reviewer. To further show of "what is VAME attaching to" we updated our generative supplemental Figure S.7 (line 994). Here, we show that VAME can reliably reproduce similar signals for a given motif from the learned latent space. Hence, it is attached to the trajectories/kinematics from which the model is learned.

The goal is to not only detect differences undetectable by human observation, but to provide scientists with a method to a) learn a motif structure that makes sense on different scales (see hierarchical tree representation in Figure 2 B) and b) to also allow studying dynamics within a lower dimensional embedding (see Figure 3 B and C).

A simple readout could not discern the two groups from each other as shown in Figure 2 A. This readout resembles exactly what the reviewer is referring to - the integral (distance travelled) of the animal nose position. Here, these simple kinematic features are shown to be not significantly different between both groups.

We moved the human classification results into the supplementals as well as the accompanying text and reduced the text about it in section 2.2 (line 134 - 137).

VAME thus offers an alternative view on behavior data - namely capturing the variance of the multivariate time series during a short time window in a single low-dimensional value. As the amount of information going into this measure is larger, it is also expected that more differences can be found. This resembles "looking at the whole body behavior" against "looking at the motion of the nose", for example. Obviously, for some situations one approach might be better than the other but we find that this analysis offers the user a more holistic view on the animal behavior and can guide the path to discovery of more fine-grained behavioral differences or interesting transition probabilities between motifs/communities.

- I may have missed it, but once trained, can this VAME model be applied to other tg/wt datasets? it seems, particularly with the small-ish n and lack of details to why a certain number of groups were selected, that this could be a problem. e.g. by random

happenstance, I can get 3 unsupported walking groups and you get 4, and the within-action difference you see may not exist in my 3.

It is true that in the current publication the proposed dataset is not very large, hence studies of generalization of the model remain a subject for future investigation. However, in ongoing different projects using VAME in our lab as well as other labs we can confirm that the model can be applied to “unseen” datasets that were not part of the training data, as long as the input time series follows approximately the same distribution. This is typically the case if several animal recordings have been captured from the same cohort, under the same circumstances, camera setup, etc. However, in general, we recommend users to include the data to be segmented into motifs also into the training dataset, as explained in the newly added protocol. If the protocol is followed, the same segmentation is applied on all animals in the datasets (tg/wt) so the clusters will correspond to the same behaviors for both groups. In that case the described scenario cannot happen as within-action differences will be focusing on the same kind of behavioral signal. On the other hand, if the model for the groups is not trained and segmented altogether, the described effect could happen. Thus, again, we advise users not to carry out the analysis in such a way.

fig 3 - I LOVED figure 3 and think we need more of it. this sort of analysis should be the future or computational neuroethology and movement study.

Thank you very much!

p7 - " On the community level, the temporal structure of behavior can be identified by observing the probability of a community transitioning into another" - It is unclear to me how this differs from saying "we can compare a walk to other walks, or to all behaviours". Apologies if I've lost the thread, but to me this is not inherent to VAME's scales or temporal structure. Most any method, even supervised, can subdivide larger communities into discrete sub-categories. If the rationale is because VAME does so in a mathematically rigorous fashion, then we now have problems with figure 4. It assumes essentially identical groupings from a qualitative/historical background. Said more simply, please help me understand why this would be a feature specific to VAME

We did not mean to create the illusion that this is a specific feature of VAME. In fact, any other method (supervised or unsupervised) can be used to subdivide communities (or cliques on a Markov graph) into discrete sub-categories. Hence, we added this point as a paragraph into the discussion (line 286 - 293):

“While VAME is effective in learning motif sequences from pose estimation signals, we also looked at the hierarchical structure of the resulting motif sequence by creating a tree representation. Within the VAME framework, motifs are sub-patterns of certain macro behaviors organized on a Markovian graph (see (Luxem et al., 2019)), Figure 3 (left)). By considering transition and usage properties on this graph, we can identify different types of e.g locomotion as shown in supplemental Figure 5.9. Hence, the tree representation transforms the motif sequence into broader, human readable categories like Walking or Rearing. This feature, though, is not unique to VAME and our approach to transform sub-patterns (or motifs) of behavior into a tree representation could be applied to any other supervised or unsupervised method.”

Fig 4 - I applaud the authors for attempting this comparison across methods. It's not a fun or comfortable thing to do, but hopefully is helpful to the field in the end. It's also difficult to compare methods given a lack of ground truth. The authors go to some effort earlier in the manuscript to point out how poorly human raters are, and that while there is no ground truth to behaviour, humans certainly shouldn't be trusted to come close to it. I would not consider 72% to be a good score for 3/3 agreement.

Thank you, the comparison of methods is something where we need to develop better metrics and benchmarks as a community in the future and I (first author) am working on this to bring people from the field together to achieve this goal. For this work we can only provide a sort-of ground truth (a true ground truth is unrealistic to achieve) given by our annotations of three experts, but more and improved metrics need to be established like a common benchmark dataset. The recent MABe 2022 challenge by Ann Kennedy and others is definitely a way forward in that sense by providing these kinds of benchmarks on multi-animal dataset, where VAME was featured as a method to build up on. We added a paragraph into our discussion section (line 314 - 322):

"In the light of our comparison between methods, we want to highlight that the future of computational ethology needs improved benchmarks and datasets for single and multi-animal behavior. Comparing methods on one particular dataset can shed some light on their performances but certain methods are better suited for certain kinds of data. With the rising amount of tools for computational ethology this becomes a pressing need and we want to trigger the development of better benchmarks and metrics that do not only rely on motif structure but also consider representations in lower dimensional space. Here, we showed that VAME is indeed capable of providing good solutions to both (see supplemental Figures 5.2 and 5.3) but we have to acknowledge that this was done in one particular setup and dataset. Future benchmarks need to evaluate these tools on much broader scenarios to come to a conclusion about when to apply a given method."

Some questions:

many details on the non-VAME methods are missing. Same DLC data used (I assume)? How many motifs were there? it seems as if there are 50 for all, which I'd assume means the other methods were forced to do so. If that is the case, this is unfair. One technique may be better at communities while another motifs - and thus we often see valid approaches with 10 vs 50 vs 100 motifs. Even if all methods aligned to 50, I'd be curious as a potential user about how the performance varied by category. This can be taken in at least 2 ways. If one technique hones in on motion energy, it may oversplit/undersplit compared to other methods that are internally consistent but don't match the human. More to the point, if I'm doing a study of rearing, I may not care about the performance in the other categories. You have plenty of text to spare. Tell us more!

Thank you, we added the details on the other models in supplemental section 5.1 (line 891 - 900) and have also now a cross-reference in section 2.4.

As stated on line 218 "We trained all models on the full dataset ...", which indicates that the same DLC data was used as for the whole manuscript.

To show the results for potential users with different numbers of k we added to Figure 4 a panel D to also exclude that the better performance of VAME is just a mere effect of a potential over clustering.

If a user is interested in only some sort of behavior, supervised methods seem to be more straightforward in finding these episodes than running a full unsupervised model.

- I'm not following something, and my apologies if that applies more broadly. This is a complicated plot, so some more explanation might be helpful. Humans rated each frame with one of 5 behavioral labels. For 4c, blue indicates how much inter-rater agreement there was on each frame that was labeled motif 0-49, yes? Either way, just a bit of text will help understanding and ultimately adoption of the technique. There is plenty of opportunity for these approaches to be unfairly applied. Provide detail and make it clear to us that this is not the case. Somewhat related, an n of 4 x 2 groups is fine, but with counts more similar to those typically used in AD mouse studies, I'd expect at least 5x that. Additionally, I'd appreciate some discussion of the empty columns. VAME has 5, while the others have 2 and 3. Could you explain for the reader what having 5 missing motifs might do to the scores in the table? Ostensibly you're comparing performance across fewer groups, which should boost VAME above the others, though it likely depends on how this is done.

We agree, the plot is a bit complicated and we apologize, but haven't found a better representation for this purpose. In 4C blue indicates the inter-rater agreement, yes. On page 9, we added additional text to clarify this (line 218 - 221).

We agree that a pure AD study would need much more animals and we acknowledged this in the discussion already (line 273 - 274):

"In our use-case, we did not aim at investigating behavioral deficits in the domain of learning and memory with relation to Alzheimer's disease."

We also updated the text in section 2.4 to address the missing columns (line 224 - 227):

"Lastly, it can be seen that columns are sometimes empty. VAME has five empty columns while the AR-HMM and MotionMapper have two and three, respectively. This can indicate that VAME is more selective about motifs and not all motifs are present in the smaller benchmark dataset (0.8% of the full dataset herein)."

p9 - " MotionMapper may be more suitable for the detection of motifs in fruit flies." I fully agree, could you please state why? I assume this comes from the reliance on motion energy. Making a clear statement about moseq would also be good here.

We discuss this point in the discussion stating (line 299 - 300) "Since the spectral energy of a signal is the key input feature, low frequency movements, which are more prominent in mice than in flies, limits capturing the full behavioral repertoire." Furthermore, we have stated about MoSeq that (line 301 - 303) "This allowed the detection of sub-second behavioral structure but the AR-HMM resulted in a multitude of short and fast switching motifs, which can lead to uncertainty in animal action classification."

A further comparison of the methods has also been included with the newly added material on genotype classification, that is now part of the supplementary material 5.1 (Model comparison) (line 911 - 920).

p9 - " All these approaches require deep, reliable and complete dissection of behavioral motifs." Absolutely! And I would LOVE to see the quantification of this in the document. This is somewhat similar to the question above about generalization, but I honestly have no idea how reliable (or complete?) VAME is. Canonical categories have multiple motifs (which is common to most if not all techniques), and VAME performs more akin to humans... though these motifs are ignored. This ignoring is reasonable, but I really don't know what the motifs mean.

We understand that the reviewer wants to see a more comprehensive study of Alzheimer's disease with VAME. Unfortunately, we cannot achieve this within this manuscript and hence, followed the advice of the reviewer to make it more akin to a methods paper by introducing the new section 4.1. But we are currently collaborating with an international group of scientists from different labs on a detailed analysis of several AD and AD-related mouse models with high n numbers, which we are convinced will show the full potential of VAME in the field of AD research. Based on this, we hope the reviewer understands that the present manuscript defines and introduces the methods, while future manuscripts use the method with focus on an AD.

p10 - " This approach, however, does not yield temporal information about the behavior" I'm not sure that I agree, at least when stated in such broad terms. I believe it uses speed (distance/time) and certainly provides the time of each action. Could the authors please clarify?

We agree with the reviewer that this statement was too vague. B-SOiD is a great tool. We updated the paragraph (line 352 - 354):

"This approach, however, does not use a trajectory sample of the behavior and projects framewise into a UMAP representation. The temporal information comes mainly from a velocity feature signal. Hence it will likely not capture the full range of behavioral dynamics."

Again, as stated above, more metrics and benchmarks are needed to really compare all of the nowadays available tools for computational ethology.

p10 / general. From the first sentence to the discussion, behavioural dynamics is mentioned, but it's not clear what the author mean. Kinematics are notably absent, and transitions can be done with any approach. I think there is something real here for the authors to capitalize on, but it's not clear to me.

We use the term behavioral dynamics freely in the manuscript as a synonym for time-dependent analysis of body part movement. The RNN model inside VAME is effectively carrying out a fit of the GRU equations to the pose estimation signal, which in a data-driven way describe the motion of the DLC markers via difference equations. Thus, the reviewer is right that we don't obtain a kinematic description of the system but a time-dependent description of the marker motion within the video frame, which we believe still resembles a

description of the dynamical movement activity. We clarified this in the Discussion section (line 258 - 261).

Minor

- page 1 - I'd assert that subsecond scale isn't the critical aspect here, and also that such an assertion is wrong (it may find larger motifs, and the eye can detect quick behaviors like tics with ease)

We agree that the eye can detect tics and other fast events with ease, but the message here is that unsupervised methods are also capable of detecting these events while supervised methods can only detect behavior that they are trained on.

- page 2 - To avoid confusion when discussing pose estimation, you may want to use "computer vision" in place of "supervised deep learning"

We only mention supervised deep learning in the context of pose estimation, which is the branch of machine learning these three methods are based on. True, the broader field is called "computer vision" as well as these tools are based on convolutional neural networks, but we don't see the confusion here.

- 1e, the text should be a bit larger. Also, if the text states 1A or 1B, the figure should not have a and b. Lastly, it looks like many things are listed as 1A but should be b,c,d...

We updated Figure 1 by increasing the text size of 1e and also updated our text by referring to it via (Figure 1 A, a), (Figure 1 A, b), etc.

- page 7 - while "Purity, Normalized Mutual Information (NMI) and Homogeneity" are all mathematically defined deep within the methods, I'm sure it would help many reader to get a handwavy explanation of these terms. Similarly, why NMI instead of AMI? Not too important, but the newer AMI metric has the benefit of being normalized to chance.

We agree with the Reviewer that we should have a handy explanation of these terms and updated section 2.4 with the following footnotes for each measure:

1. Purity is a measure of the extent to which clusters contain a single class.
2. From scikit-learn: Normalized Mutual Information (NMI) is a normalization of the Mutual Information score to scale the results between 0 (no mutual information) and 1 (perfect correlation).
3. Homogeneity is in its essence a more strict Purity measure. From scikit-learn: A clustering result satisfies homogeneity if all of its clusters contain only data points which are members of a single class."

While we agree that AMI has some benefits to it, we implemented the score and found that the figures were very similar to the NMI ones in our case. We thus decided to report the NMI score but will consider using the AMI in future studies.

- page 7 - I'd change " In order to validate all three models (VAME, AR-HMM, MotionMapper) we created..." to "To validate three models (VAME, AR-HMM used in MoSeq, MotionMapper) we created..."

We have changed it accordingly.

- I appreciate the correct spelling of behaviour in some parts of the manuscript, but it is inconsistent.

We agree that we should be consistent about this. Based on the fact that our title uses the spelling "behavior" we decided to use this version throughout the manuscript.

Reviewer #4 (Remarks to the Author):

This excellent submission details a new framework (VAME) for identifying hierarchical structure from recordings of animal behavior. The authors demonstrate the utility of their approach by identifying (biologically relevant) differences between transgenic and wild-type animals, and show that VAME performs favorably on a benchmark dataset when compared to similar approaches.

I am convinced that the paper will be of substantial interest to the community; particularly given the exceptionally well documented code provided on github. Indeed, the communities' interest is already clear from the > 30 citations the associated preprint has received and almost 100 stars on github. Relatedly, by providing high quality code and detailed parameter choices (Table S.3) the authors maximize the potential reproducibility of their work.

Thank you a lot for these kind words, we really appreciate it!

I would, however, like to suggest a number of minor changes which would strengthen the manuscript.

Minor Changes:

1. In section 2.1 it is not clear how "six virtual markers" (4 paws, nose, tailbase) yield "10 (x, y) - marker positions".

We addressed this by updating our method section 4.3 (line 603 - 606) and having now a reference to this section in section 2.1. This is a result of the egocentric alignment.

2. In section 2.1 and the first paragraph of the discussion the authors the phrase "richer representation", which is vague. The authors should describe the difference in the representation with and without the additional biRNN decoder, and ideally add results to the supplement detailing this comparison.

We agree with the reviewer that the phrase "richer representation" is indeed vague and changed it into "forcing the encoder to learn improved dynamical features from the behavioral time series" in section 2.1 (line 93 - 94). Moreover, we highlight in section 2.1 our

comparison from section 5.3, where we test the performance of a single or bi-decoder network. Within the discussion part, we changed the phrase “richer representation” as well to “improved dynamical features from the behavioral time series” (line 258).

3. While some choices regarding hyperparameters are always necessary, it would be nice if the authors could better justify their choice of 30 dimensions for their latent representation (section 2.2).

Indeed, the choice of the number of latent dimensions is critical to the VAE framework. As described in 2.2 the number of 30 dimensions was empirically set by inspecting motif video representation and reconstruction score. However, we agree with the reviewer that the choice of the number of latent dimensions should be estimated based on a goodness metric. Based on the full review, we added a new section 4.1 (from line 373) to the manuscript to guide new and advanced users through our method. More precisely, in section 4.1.4 we added the following (line 481 - 485):

“The other parameter, z_{dims} , ensures that the model embeds a lower dimensional representation of the input trajectory. To identify a good setting of this parameter, users could train their model with different settings and evaluate when the reconstruction and prediction mean-squared-error losses plateau. For the demonstration data we found that the plateau started at around 12 latent dimensions (see supplemental Figure S.13).”

We are also discussing the number of latent dimensions now in the discussion (line 323 - 332).

4. Related to point 3, the authors should discuss if the same latent dimension size is appropriate for different animals or for animals with different genetic backgrounds (i.e., wild-type / transgenic).

We agree with the reviewer that this number needs to be chosen carefully. We added the following paragraph to the discussion to make the reader more aware of this problem (line 323 - 332): “An important aspect of VAME are the choices of its hyperparameter, which we summarized in table S.3. Here, the number of latent dimensions is one of the central parameters. The embedded dimension controls the amount of information the model can extract. Based on the information bottleneck theory, this number should be chosen as small as possible to extract the most salient information from the data. However, this is coupled with multiple factors. One of the principal factors for VAME are the choice of time window w and number of marker coordinates m as this is the information VAME is condensing into a vector representation. Higher numbers of w or m (or both) might result either into needing more latent dimensions or into pre-processing the data through a top layer neural network or with other techniques like principal component analysis. Users of VAME need to adjust this number to their needs and should be aware that this number significantly affects the outcome of their VAME model. Having a benchmark dataset and investigating the reconstruction score can help to identify an appropriate number.”

5. In section 2.2, the authors should make it clearer that they identify the same 50 motifs in each animal (as opposed to 50 unique motifs per animal or a mixture of shared and unique motifs).

We updated section 2.2 to make it clearer that the same 50 motifs are identified for each animal (line 143 - 146) "... we inferred the same 50 motifs per animal (see section 5.5 for details) to be able to compare behavioral structure between groups. We then created a hierarchical tree representation C from the motif structure of the full cohort of animals ... "

6. In section 2.2 the authors should highlight that motif usage is highly similar (has low variance) across animals. This would support their claim that VAME identifies robust / common behaviors.

We agree with the reviewer and highlighted this in section 2.2 by adding (line 161 - 162) "Most motifs showed a low variance in usage between animals for a given group but there also are motifs where differences between groups are visually apparent (Figure 2, C)."

Moreover, we added a sentence to the discussion part (line 277 - 278) "Interestingly, the motif usage within groups showed a low variance, which points towards the robustness of the method to detect a common and stereotyped behavioral structure."

7. However, following point 6, it is not clear to me if the author's approach would be able to identify motifs used by only one population (here half of the animals). Such motifs would be of interest as they would represent significant changes in behavior.

We understand the concern of the reviewer. Indeed, detecting motifs that are present in one population but not in the other would be of special interest as they show a completely different set of behavior. We added this to the discussion (line 333 - 346):

"Furthermore, choosing the number of appropriate motifs is another question, which is hard to generalize, as every experiment and/or animal used will have their unique set of behaviors and hence number of motifs. In this work, we considered only motifs that have a higher than 1% usage after sampling 100 motifs from our embedding space and re-run the motif segmentation with this number. In general, however, it would be of special interest to identify motifs that are present in one group/animal but not the other. This would show a complete different set of behavior and mark highly significant differences between them. Our data is very homogeneous in terms of behavior and the two populations do not differ drastically in their executed behavior. Hence, such motifs are not likely to appear in this work. However, we believe that VAME is capable of finding these "out-of-distribution" motifs when they exist based on the variational autoencoding framework. In general, for every behavioral quantification method there is a trade-off of how general the motif distribution should be versus how precise individual behavior is measured. If the goal is to identify very specialized individual behavior it would be possible to parameterize both populations or all animals individually. The problem lies in relating motifs with each other between populations/animals as the motif mapping would change per parameterization. Users of these tools should keep this in mind and this question needs to be addressed by future work."

8. Figure S.1 (particularly panels A, B and C) provide an intuitive graphical description of the author's comparison between wild-type and transgenic animals. The manuscript would be improved if the authors could either incorporate some aspects of S.1 into Figure 2, or swap table 1 and S.1. I acknowledge that this may be preference of style.

We appreciate the comment of the reviewer and changed the style of Figure 2 by including panel A, B and C from Figure S.1. This, indeed, provides a more intuitive description of the comparison between both groups. We also updated the main text of section 2.2 and updated Figure S.4 with the remaining parts from the old Figure 2.

9. In Figure 2B the authors convincingly demonstrate that human observers cannot classify wild-type vs transgenic animals with greater than chance accuracy. The authors should extend this result by showing that VAME (presumably) enables greater classification accuracy based on motif usage. One approach to this would be to use linear discriminant analysis classifiers. In Ghosh and Rihel 2020 (eNeuro), for example, the authors demonstrate that differences in motif usage can be used to accurately distinguish larval zebrafish behavior across a variety of contexts, such as exposure to different concentrations of a compound (Figure 5A and B).

While we think that the phenotype separation task can be misleading given the very homogeneous behavior of the animals tested in this study, we still went ahead and performed LDA classification with the motif distribution from VAME, AR-HMM and MotionMapper. The outcome was that all models can separate the phenotypes to a certain degree meaning that each model had a mixed-up animal (always a different one). We added the results to our supplemental section 5.1 (Model comparison), where we show in Figure S.2 the classification probability of each model and the following text (line 911 - 920):

“To further compare VAME, AR-HMM and MotionMapper, we tested their ability to separate phenotypes based on their motif distribution. Here, we used the same approach as in (M. Ghosh & Rihel, 2020). Briefly, the authors used a linear discriminant analysis (LDA) classifier to demonstrate that differences in motif usage can be used to accurately distinguish larval zebrafish behavior across a variety of contexts, such as exposure to different concentrations of a compound. We applied the same classifier to the motif distribution output of the models tested herein. The results show that all three models can separate the phenotype to a certain degree meaning that each model has an animal misclassified. One obvious reason for this could be the strong homogeneity in behavior between the two groups. Supplemental Figure S.2 shows the LDA classification probability for each model. Given our small group size this can only be interpreted as a trajectory of what these models are capable of with higher numbers of a certain phenotype and hence makes them ideal candidates to study behavior in disease animal models.”

10. Related to 9, a comparison of the motif usage-based classification accuracy (wild-type vs transgenic) across VAME, AR-HMM and MotionMapper would be an interesting addition to the paper, as it would determine which method is best able to detect a biologically relevant difference.

See answer to comment 9.

11. The comparison between manual annotators and methods is convincing (Table 2), however a graph of these values as a function of k would strengthen the authors claim.

We agree with the reviewer and added a panel D to our main Figure 4, where we show the

performance of all three methods (VAME, AR-HMM, and MotionMapper) as function of k for all three metrics. We also updated the main text in section 2.4 (line 231 - 233) "In Figure 4 D, we further showed that VAME achieves the best scores on all three metrics when measured as a function of motif number k. Interestingly, the performance of VAME stays stable even for small motif numbers compared to the AR-HMM and MotionMapper."

12. In the discussion the examples of supervised approaches are repeated: "Supervised approaches like DeepEthogram, SimBa or MARS DeepEthogram, SimBa or MARS".

We fixed the typo in the discussion (line 349).

13. In general it would be good if the authors could better motivate their use of a bottom-up camera, and discuss what one may hope to discover from applying VAME to existing datasets acquired from this perspective. For example, Darmohray ... and Carey 2019 (Neuron).

We updated section 2.1 and motivated the bottom-up camera usage more explicitly (line 70 - 73): "A major advantage of the bottom-up perspective is that it reveals most of the animals kinematic with only one camera, which can be efficiently tracked by pose estimation. Our goal is to build a model that can learn the behavioral structure purely from the kinematic pose tracking."

14. The general appeal of the paper would be increased if the authors could expand upon how VAME could be applied to other types of data.

Within our new section 4.1 we expand upon this by adding the following text (line 380 - 383): "VAME is a general time series quantification method and while we used in our exemplary data pose tracking input from DeepLabCut, VAME works also with other pose estimation tools like SLEAP, DeepPoseKit or B-KinD (T. D. Pereira et al., 2022; Graving et al., 2019; Sun, Ryou, et al., 2021). In principle, other kinds of data such as a principal component time series of the video data or other sensory signals can be fed into the model."

15. Finally, the text accompanying supplement 5.8 (Generative model aspects) would be improved with more interpretation of the figure. Currently, it is not clear what the take away is, or more precisely how similar the reconstructions are to the generated samples.

We agree with the reviewer and updated the supplemental section 5.8. We now show in Figure S.7 (A-C) three example motifs and generate from their latent distribution the DeepLabCut trajectories. We added the following text (line 1004 - 1008): "In Figure S.7 (A-C), we use three exemplary motifs and sample from their latent distribution. By using the reconstruction decoder of VAME as a generative model, we can show that the distributions are well defined since the decoder is reconstructing very similar trajectory samples. Note that we did not use any input sample to decode the motif trajectories and this is all coming from the learned latent distribution. Figure S.7 (D) shows trajectory samples form randomly sampled data points of our latent distribution"

REVIEWERS' COMMENTS:

Reviewer #1 (Remarks to the Author):

The authors have addressed my concerns.

Reviewer #2 (Remarks to the Author):

Thank you for the excellent clarification in response to all the comments. There were several issues to be address at the request of the reviewers, and the authors have done so. I am very supportive of the current manuscript, especially with the following small comments.

1- The authors' discussion about generalization of the method was excellent. I would ask that they include some or part of their reply in the actual text.

2- "The temporal information comes mainly from a velocity feature signal. Hence it will likely not capture the full range of behavioral dynamics."

I don't know of any real evidence or theory to support the second sentence, and being real doesn't hurt or help VAME. Unless the authors can demonstrate something to back up the idea, I would remove the second sentence as hearsay.

3- double check for typos, e.g. 'Figure 4 C' and 'Figure 4, C' in lines 196, 198; line 324 is instead of are.

Reviewer #3 (Remarks to the Author):

I was pleased to receive the revised version of this excellent manuscript.

The authors have addressed all of my own points and, from reading the other reviews and comments, those of my colleagues. In my opinion the manuscript has been improved by these changes. In particular, the new section (4.1), a detailed guide on how to setup, use and interpret VAME, is a great addition and I am sure that it will enhance the usage and reproducibility of their work.

I look forward to seeing work which builds on this paper, both from these authors and the community.

Point-to-Point response letter

We thank all three reviewers for their time and work invested into reviewing our manuscript.

Below we address each point raised by the respected reviewer and highlight changes within our manuscript.

Reviewer #1 (Remarks to the Author):

The authors have addressed my concerns.

Thank you!

Reviewer #2 (Remarks to the Author):

Thank you for the excellent clarification in response to all the comments. There were several issues to be address at the request of the reviewers, and the authors have done so. I am very supportive of the current manuscript, especially with the following small comments.

Thank you!

1- The authors' discussion about generalization of the method was excellent. I would ask that they include some or part of their reply in the actual text.

We included the discussion about generalization as a paragraph within the Discussion section on page 7. We wrote:

“Applying a trained VAME model to other unseen animal datasets can be beneficial in terms of reducing computational costs and identifying similar motif distributions. A caveat, however, lies in the fact that the unseen data must follow approximately the same data distribution as the training and test set. This is typically the case if several animal recordings have been captured from the same cohort, under the same circumstances, camera setup, etc. However, in general, we recommend users to include the data to be segmented into motifs also into the training dataset, as detailed within the Methods.”

2- “The temporal information comes mainly from a velocity feature signal. Hence it will likely not capture the full range of behavioral dynamics.”

I don't know of any real evidence or theory to support the second sentence, and being real doesn't hurt or help VAME. Unless the authors can demonstrate something to back up the idea, I would remove the second sentence as hearsay.

We agree with the reviewer and removed the second sentence here.

3- double check for typos, e.g. 'Figure 4 C' and 'Figure 4, C' in lines 196, 198; line 324 is instead of are.

We cleaned up these typos.

Reviewer #3 (Remarks to the Author):

I was pleased to receive the revised version of this excellent manuscript.

The authors have addressed all of my own points and, from reading the other reviews and comments, those of my colleagues. In my opinion the manuscript has been improved by these changes. In particular, the new section (4.1), a detailed guide on how to setup, use and interpret VAME, is a great addition and I am sure that it will enhance the usage and reproducibility of their work.

I look forward to seeing work which builds on this paper, both from these authors and the community.

Thank you!